# Meta-ethnography to understand healthcare professionals' experience of treating adults with chronic non-malignant pain

Francine Toye,[1] Kate Seers,[2] Karen L Barker[3]

[1]Nuffield Orthopaedic Centre, Oxford University Hospitals NHS Foundation Trust, Oxford, UK
[2]Royal College of Nursing Research Institute, Warwick Medical School, University of Warwick, Coventry, UK
[3]Nuffield Department of Orthopaedics, Rheumatology and Musculoskeletal Sciences, University of Oxford, Oxford, UK

**Correspondence to**
Dr Francine Toye;
francine.toye@ouh.nhs

## ABSTRACT

**Objectives** We aimed to explore healthcare professionals' experience of treating chronic non-malignant pain by conducting a qualitative evidence synthesis. Understanding this experience from the perspective of healthcare professionals will contribute to improvements in the provision of care.

**Design** Qualitative evidence synthesis using meta-ethnography. We searched five electronic bibliographic databases from inception to November 2016. We included studies that explore healthcare professionals' experience of treating adults with chronic non-malignant pain. We used the GRADE-CERQual framework to rate confidence in review findings.

**Results** We screened the 954 abstracts and 184 full texts and included 77 published studies reporting the experiences of over 1551 international healthcare professionals including doctors, nurses and other health professionals. We abstracted six themes: (1) a sceptical cultural lens, (2) navigating juxtaposed models of medicine, (3) navigating the geography between patient and clinician, (4) challenge of dual advocacy, (5) personal costs and (6) the craft of pain management. We rated confidence in review findings as moderate to high.

**Conclusions** This is the first qualitative evidence synthesis of healthcare professionals' experiences of treating people with chronic non-malignant pain. We have presented a model that we developed to help healthcare professionals to understand, think about and modify their experiences of treating patients with chronic pain. Our findings highlight scepticism about chronic pain that might explain why patients feel they are not believed. Findings also indicate a dualism in the biopsychosocial model and the complexity of navigating therapeutic relationships. Our model may be transferable to other patient groups or situations.

## BACKGROUND

Chronic pain is defined as pain that persists beyond the 3 months expected time of healing.[1] In 2009, an estimated 5 million people in the UK developed chronic pain,[2] and a recent systematic review suggests that this may underestimate the problem.[3] Around 20% of adults in Europe have

### Strengths and limitations of this study

► This study brings together, for the first time, a large number of qualitative studies (n=77) that explore the experience of healthcare professionals' experience of treating people with chronic non-malignant pain.
► Meta-ethnography provides the reviewers' interpretation of qualitative findings abstracted into a line of argument with the aim of providing food for thought.
► There is no consensus on how to assess the quality of primary qualitative studies.
► Although the GRADE-CERQual provides a useful framework for determining confidence in qualitative syntheses, there is currently no consensus on how to do this.

chronic pain,[4] and in the USA, more than 25 million adults (11%) experience chronic pain.[5] Chronic pain is challenging, because it persists beyond healing time and is not easy to explain or treat. A range of clinical staff are involved in caring for people with chronic pain, and in the UK, there is a wide range in the provision of specialist care.[6] Not all patients with chronic pain have access to specialist services, and a national UK audit in 2012 indicated that only 40% of pain clinics met the minimum standard of having a psychologist, physiotherapist and physician.[6] The audit suggests that as many as 20% of patients with chronic pain visit accident and emergency departments even after visiting their general practitioner (GP), and as many as 66% visit a clinician three times within a 6-month period. A survey of undergraduate pain curricula for healthcare professionals (HCPs) in the UK[7] indicates that although these curricula are available, pain education is highly variable and 'woefully inadequate given the prevalence and burden of pain'[7] (p. 78).

The Cochrane Qualitative Research Methods Group acknowledges the importance of including qualitative findings within evidence-based healthcare.[8] Qualitative evidence synthesis (QES) aims to bring together qualitative research findings to make them accessible for public, policy, practice and education. A recent synthesis of 11 QES has highlighted the personal challenge of living with chronic non-malignant pain and the loss of personal credibility that is integral to this experience.[9] Findings from a QES of 77 qualitative studies exploring patients experience of living with chronic non-malignant pain also demonstrate that patients can experience healthcare as an adversarial battle.[10] Understanding this from the perspective of HCPs will help us to unpick this experience and thus contribute to improvements in care provision. Although there is a large body of qualitative research exploring HCPs experience of treating chronic non-malignant pain, there has been no attempt to systematically search for and integrate this knowledge into a QES. We aimed to conduct a QES using the methods of meta-ethnography.[11] Meta-ethnography is widely used and has provided insight into healthcare experiences such as medicine taking,[12] diabetes[13] antidepressants,[14] osteoporosis,[15] chronic musculoskeletal pain[10 16] and chronic pelvic pain.[17]

## METHODS

Meta-ethnography is a method developed by Noblit and Hare that aims to synthesise qualitative research findings into a whole that is greater than the sum of its original parts.[11] We used the methods of meta-ethnography developed, refined and reported by Toye and colleagues.[10 18] There are various methods for synthesising qualitative research.[19–23] An important distinction is between (A) those that *describe* findings and (B) those, like meta-ethnography, that develop *conceptual* understandings through a process of constant comparison and abstraction.[11] There are seven stages to meta-ethnography: getting started, deciding what is relevant, reading the studies, determining how studies are related, translating studies into each other, synthesising translations and expressing the synthesis.[11]

In their original text, Noblit and Hare do not advocate an exhaustive search,[11] and the number of studies included in meta-ethnographies ranges.[20 22 24] Unlike quantitative syntheses, qualitative syntheses do not aim to summarise the entire body of available knowledge or make statistical inference. We searched five electronic bibliographic databases (MEDLINE, Embase, Cinahl, PsycINFO and Amed) using terms adapted from the InterTASC Information Specialists' Sub-Group Search Filter Resources.[25–28] We used subject headings and free-text terms for *qualitative research*, combined with subject heading and free text terms for *pain* (table 1). We did not include citation checks, hand searching, grey literature or PhDs as, in our previous QES, 95% of the included studies were identified in the first three databases searched.[10]

We included studies that explored the experience of all clinical healthcare staff involved in the care of patients with chronic pain. We excluded: acute pain, head pain and arthritic conditions. FT and KLB screened the titles, abstracts and full text of potential studies.

There is currently no consensus on what makes a qualitative study *good enough* for QES.[24 29] However, a growing number of reviewers are appraising studies for QES.[22] We did not intend to use rigid guidelines but felt it important to seriously consider quality. We used three methods of appraisal: (A) The Critical Appraisal Skills Programme (CASP) questions for qualitative research[30]; (B) constructs from a qualitative study in a previous meta-ethnography[31] and (C) a global appraisal of whether the study was: 'key' (conceptually rich), 'satisfactory', 'irrelevant' or 'fatally flawed'.[29] As some journals are not explicit about ethical approval, we screened potential studies for ethical standards (CASP question 7: have ethical issues been taken into consideration?[30]). If FT and KLB did not agree about inclusion, they consulted KS for a final decision. We used the GRADE-CERQual framework,[32] which aims to rate how much confidence readers can place in review findings. GRADE-CERQual suggest four domains: (1) '*Methodological limitations*'; (2) '*Relevance*'; (3) '*Adequacy of data*' (the 'degree of richness and quantity of data supporting a review finding'); (4) '*Coherence*' (consistency across primary studies), and finally, an overall rating of confidence (high, moderate, low and very low).

We planned to develop a line of argument synthesis, which involves 'making a whole into something more than the parts alone imply'[11] (p. 28). Analysis in large QES involves a process of: identifying concepts from qualitative studies, abstracting these concepts into conceptual categories, further abstracting categories into themes and finally developing a line of argument that makes sense of the themes. We read studies in batches of topic or professional grouping. We did not use an *index* paper to *orientate* the synthesis,[33] as we felt that this choice can have a dramatic impact on the interpretation.[18] Two reviewers read each paper to identify, describe and list concepts. If they agreed that there was no clear concept, then it was excluded. Through constantly comparing and discussing concepts, three reviewers abstracted concepts into conceptual categories, using NVivo V.11 software for qualitative analysis to keep track of our analytical decisions.[34] NVivo is particularly useful for collaborative analysis as it allows the team to keep a record and compare interpretations. Once we had agreed and defined conceptual categories, these were printed onto cards and sent to our advisory group to read and sort into thematic groups. This group consisted of patients, allied health professionals, nursing professionals, doctors and managers. Then, during advisory meeting, the reviewers worked alongside the advisory group to finalise the themes that would be included in the line of argument. In this way, we were able to challenge our own interpretations. Some reviewers do not present a line of argument as part of their QES findings. Frost and colleagues indicate that there has been a move away from

| Table 1 | Example search syntax for MEDLINE |
|---|---|
| (I) Qualitative subject headings | EXP QUALITATIVE RESEARCH<br>EXP INTERVIEWS AS TOPIC<br>EXP FOCUS GROUPS<br>NURSING METHODOLOGY RESEARCH<br>ATTITUDE TO HEALTH |
| (II) Qualitative free-text terms | qualitative adj5 (theor* or study or studies or research or analysis)<br>ethno.ti,ab<br>emic or etic. ti,ab<br>phenomenolog*.ti,ab<br>hermeneutic*.ti,ab<br>heidegger* or husserl* or colaizzi* or giorgi* or glaser or strauss or (van and kaam*) or (van and manen) or ricoeur or spiegelberg* or merleau).ti,ab<br>constant adj3 compar*.ti,ab<br>focus adj3 group*.ti,ab<br>grounded adj3 (theor* or study or studies or research or analysis).ti,ab<br>narrative adj3 analysis.ti,ab<br>discourse adj3 analysis.ti,ab<br>(lived or life) adj3 experience*.ti,ab<br>(theoretical or purposive) adj3 sampl*.ti,ab<br>(field adj note*) or (field adj record*) or fieldnote*.ti,ab<br>participant* adj3 observ*.ti,ab<br>action adj research.ti,ab<br>(digital adj record) or audiorecord* or taperecord* or videorecord* or videotap*).ti,ab<br>(cooperative and inquir*) or (co and operative and inquir*) or (co-operative and inquir*).ti,ab<br>(semi-structured or semistructured or unstructured or structured) adj3 interview*.ti,ab<br>(informal or in-depth or indepth or 'in depth') adj3 interview*.ti,ab<br>(face-to-face' or 'face to face') adj3 interview*.ti,ab<br>'ipa' or 'interpretative phenomenological analysis'.ti,ab<br>'appreciative inquiry'.ti,ab<br>(social and construct*) or (postmodern* or post-structural*) or (post structural* or poststructural*) or (post modern*) or post-modern* or feminis*.).ti,ab<br>humanistic or existential or experiential.ti,ab |
| (III) Pain subject headings | EXP BACK PAIN/OR EXP CHRONIC PAIN/OR EXP LOW BACK PAIN/OR EXP MUSCULOSKELETAL PAIN/OR EXP PAIN/OR EXP PAIN CLINICS/.<br>EXP FIBROMYALGIA/<br>EXP PAIN MANAGEMENT/ |
| (IV) Pain free-text terms | (chronic* or persistent* or long-stand* or longstand* or unexplain* or un-explain*) fibromyalgia<br>'back ache' or back-ache or backache<br>'pain clinic' or pain-clinic*<br>pain adj5 syndrome* |

interpretation and theory development in QES towards aggregative forms of synthesis.[35] The final analytic stage, 'synthesising translations', involved the three reviewers working together alongside the advisory group to craft the final themes into a 'line of argument' to build up a picture of the whole, grounded in the themes.

## FINDINGS

We retrieved 184 full texts and excluded 101 studies (figure 1). We excluded 16 studies that were not qualitative or that included limited qualitative data.[36–51] We agreed that 85 studies were out of scope[52–136] (eg, they did not present the HCP voice or they did not explore the experience of chronic pain). Of the 83 studies remaining, we unanimously excluded six on the grounds of methodological report.[137–142] FT and KLB unanimously appraised

five studies as 'key papers'[143–147] and the remaining studies were appraised as 'satisfactory'. They did not agree about four studies[148–151] that were subsequently included. We included 77 published studies .[143 144 146–219] THIS SHOULD BE REFS 143-219 please add 145 reporting the experiences of over 1551 HCPs from USA (20 studies), UK (18 studies), Canada (10 studies), Sweden (10 studies), the Netherlands (4 studies), Norway (4 studies), Australia (4 studies), France (4 studies), Germany (1 study), Hong Kong (1 study), Ireland (1 study), Israel (1 study), Italy (1 study) and Spain (1 study) (table 2). We agreed that ethical issues had been satisfactorily considered in the study design of all 77 studies, and none were excluded on ethical grounds. Six studies were published before 2000, 37 were published between 2000 and 2010 and 34 were published from 2011 onwards. HCPs included

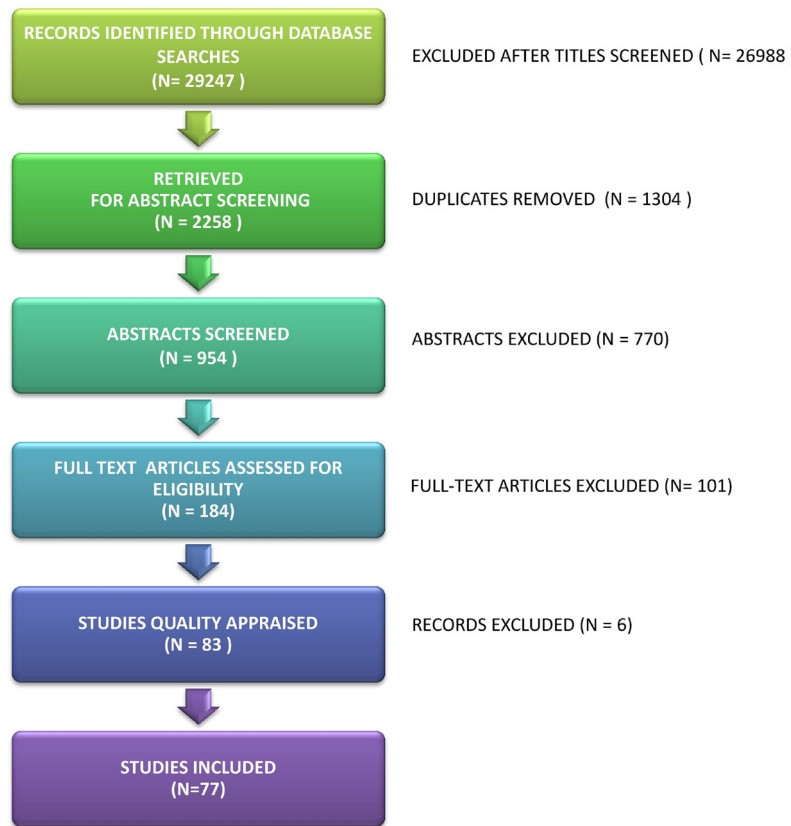

**RECORDS IDENTIFIED THROUGH DATABASE SEARCHES**
(N= 29247 )

EXCLUDED AFTER TITLES SCREENED ( N= 26988 )

**RETRIEVED FOR ABSTRACT SCREENING**
(N = 2258 )

DUPLICATES REMOVED  (N = 1304 )

**ABSTRACTS SCREENED**
(N = 954 )

ABSTRACTS EXCLUDED (N = 770)

**FULL TEXT  ARTICLES ASSESSED FOR ELIGIBILITY**
(N = 184)

FULL-TEXT ARTICLES EXCLUDED (N= 101)

**STUDIES QUALITY APPRAISED**
(N = 83 )

RECORDS EXCLUDED (N = 6)

**STUDIES INCLUDED**
(N=77)

**Figure 1** Flow diagram of records identified and studies removed and included.

doctors, nurses and other health professionals in various contexts and geographical locations. Not all of the studies reported the number of participants from specific professional groups, which means that it was not possible to give the exact number of participants from each profession. Table 2 shows the author, year of publication, country, sample size, data collection, analytic approach, professional group/context, participants and study focus. The studies explored the experience of: GPs (10 studies); mixed HCPs in diverse contexts (4 studies); physiotherapists (3 studies); physiotherapists with a specialty in chronic pain (3 studies); mixed HCPs in fibromyalgia (5 studies); mixed HCPs in chronic pain services (11 studies); mixed HCPs in pain management related to employment (5 studies); mixed HCPs prescribing opioids to patients with chronic pain (12 studies); mixed HCPs using guidelines for chronic pain (6 studies); mixed HCPs working with older adults with chronic pain (3 studies); mixed HCPs working in long term care facilities (13 studies); and nurses (2 studies).

Two reviewers identified 371 concepts from the 77 studies included. They organised the 371 concepts into 42 conceptual categories and then into six themes: 15 out of 371 concepts did not fit our analysis (supplementary appendix 1). There were several topics with insufficient weight to develop robust themes: ethnicity,[172 201 202 211 212] gender[203] and older people.[160 161 163 166 176 186 193 197 205] These may indicate useful areas of further research. Experience specific to opioid prescribing is reported elsewhere. A short

film presenting the key themes is available on YouTube (https://www.youtube.com/watch?v=477yTJPg10o)  and a report giving further details of analytical decisions is being published by the NIHR Journals library https://www.journalslibrary.nihr.ac.uk/programmes/hsdr/1419807/#/.

The six final themes were: (1) a sceptical cultural lens; (2) navigating juxtaposed models of medicine; (3) navigating the geography between patient and clinician; (4) the challenge of dual advocacy; (5) personal cost; and (6) the craft of pain management. These themes are illustrated below with narrative exemplars. Indicators of confidence in each review findings are shown in table 3, which shows: the number of studies rated as key/valuable or satisfactory (*methodological limitations*); the number of concepts (*adequacy*); the number of studies out of 77 (*coherence*); an assessment of study relevance; and our overall assessment of confidence. We rated our confidence in the review finding as high when it was supported by more than half of the studies (n≥39). However, there is currently no agreed way of making an assessment of confidence for QES. We aimed to explore HCPs' experience of treating chronic non-malignant pain. We found that studies explored the experiences of diagnosing *and* treating chronic pain and that these experiences were inextricably linked. The studies supporting each theme are shown in table 3. The themes are drawn from a wide range of HCPs, including those specialising in chronic pain management

**Table 2** Author and year of publication, country, data collection method, analytic approach, order of analysis and professional group and context, participants and study focus

| Author, year | Country | Data collection | Analytical approach | Order of analysis and professional group/context | Participants | Study focus |
|---|---|---|---|---|---|---|
| Afrell, 2010[146] | Norway | Focus groups | Phenomenology | 4. Specialist physiotherapists | 6 Physiotherapists with 10–15 years' experience (three pain management, two primary care, one private practise) | To explore physiotherapists experiences of using key questions when assesing patients with long-standing pain |
| Allegretti, 2010[152] | USA | Semistructured interview | Immersion–crystallisation | 1. Primary care physicians/GPs | 13 Physicians (five residents eight attending) | To explore shared experiences among chronic LBP patients and their physicians |
| Asbring, 2003[153] | Sweden | Semistructured interview | Grounded theory | 5. Fibromyalgia | 26 Physicians (GP, rheumatology, infectious diseases, rehabilitation, internal medicine and neurology) | To explore: (1) how physicians describe patients with chronic fatigue and fibromyalgia; (2) what the conditions mean to physicians; (3) strategies used |
| Baldacchino, 2010[154] | Scotland | Focus groups and interviews | Framework analysis | 8. Opioid prescription | 29 Physicians (primary care, addiction specialists, pain specialist, rheumatologist) | To explore physicians' attitudes and experience of prescribing opioids for chronic non-cancer pain with a history of substance abuse |
| Barker, 2015[155] | UK | Semistructured interviews | Action research | 4. Specialist physiotherapists | 7 Physiotherapists (one clinical lead, three advanced practitioners, two senior physiotherapists, one assistant practitioner) | To explore the implementation of Acceptance and Commitment Therapy to physiotherapy-led pain rehabilitation programme |
| Barry, 2010[156] | USA | Semistructured interview | Grounded theory | 8. Opioid prescription | 23 Physicians | To explore physicians' attitudes and experiences about treating chronic non-cancer pain |
| Baszanger, 1992[157] | France | Ethnography | Grounded theory | 6. Chronic pain services | NK physicians: 326 consultations of pain medicine specialists observed | To explore how physicians specialising in pain medicine work at deciphering chronic pain |
| Berg, 2009[158] | USA | Semistructured interview | Thematic analysis | 8. Opioid prescription | 12 Physicians 4 'Physician assistants' | To explore providers' perceptions of ambiguity and examine strategies for making diagnostic and treatment decisions to manage chronic pain (methadone maintenance therapy) |
| Bergman, 2013[159] | USA | Interviews | Thematic analysis | 1. Primary care physicians/GPs | 14 GPs | To explore the experiences of patients and HCPs communicating with each other about pain management in the primary care setting |
| Blomberg, 2008[160] | Sweden | Focus groups | Grounded theory | 11. Nursing | 20 District nurses (10 with pain management training) | To explore and explain district nurses care of chronic pain sufferers |
| Blomqvist, 2003[161] | Sweden | Interviews | Content analysis | 9. Older adults | 52 Mixed HCPs (35 auxiliary nurses, 13 registered nurses, 4 PT/OTs | To explore HCPs' perceptions of older people in persistent pain |

Continued

**Table 2** Continued

| Author, year | Country | Data collection | Analytical approach | Order of analysis and professional group/context | Participants | Study focus |
|---|---|---|---|---|---|---|
| Briones-Vozmediano, 2013[162] | Spain | Semistructured interview | Discourse analysis | 5. fibromyalgia | **9 Mixed HCPs** (GPs, occupational health physicians, physiotherapists, rheumatologists, psychologists, psychiatrist) | To explore experiences of fibromyalgia management, diagnostic approach, therapeutic management and the health professional–patient relationship |
| Cameron, 2015[163] | Scotland | Semistructured telephone interviews | Thematic analysis | 9. Older adults | **13 Mixed HCPs** (GPs, anaesthetist, elderly care physician, OT/PT, nurse, psychologist) | To explore attitudes and approaches to pain management of older adults, from the perspectives of HCPs' representing multidisciplinary teams |
| Cartmill, 2011[164] | Canada | Semistructured interview | Grounded theory | 6. Chronic pain services | **10 Mixed HCPs** (OT/PT, kinesiology, cognitive behavioural, psychology, work, resource specialty, customer service) | To explore the experience of transition from an interdisciplinary team to a transdisciplinary model of care in a functional restoration programme for chronic MSK pain |
| Chew-Graham, 1999[165] | UK | Semistructured interview | Grounded theory | 1. Primary care physicians/GPs | **20 GPs** | To explore how GPs understand chronic low back pain, how they approach the consultation and how they conceptualise the management of this problem |
| Clark, 2004[166] | USA | Focus groups | Ethnography | 10. Pain in age care facilities | **NK mixed** (licenced and unlicenced care nurses and other workers) | To explore nursing home staff's pain assessments in nursing home residents |
| Clark, 2006[167] | USA | Semistructured interviews | Content analysis | 10. Pain in age care facilities | **103 Mixed HCPs** (9 administrators, 38 registered nurses, 26 licenced practical nurses, 22 certified nursing assistants, 2 rehab therapists, 3 social workers, 3 directors/assistants) | To explore the perceptions of a nursing home staff who participated in a study to develop and evaluate a multifaceted pain management intervention |
| Côté, 2001[168] | Canada | Focus groups | Thematic analysis | 7. Pain-related work disability | **30 Chiropractors** (involved in return to work) | To explore concept of timely return-to-work with musculoskeletal injuries, approaches to treatment of injured workers and perspectives on the barriers and facilitators of successful return-to-work |
| Coutu, 2013[169] | Canada | Semistructured interviews | Thematic analysis | 7. Pain-related work disability | **5 Occupational therapy/kinesiology** | To explore differences between clinical judgement, workers' representations about their disability and clinicians' interpretations of these representations |
| Dahan, 2007[170] | Israel | Focus groups | Immersion–crystallisation | 9. Guidelines | **38 GPs** | To explore barriers and facilitators for the implementation of low back pain guidelines from GPs' perspective |
| Daykin, 2004[171] | UK | Semistructured interviews | Grounded theory | 3. Physiotherapists | **6 Physiotherapists** | To explore physiotherapists' pain beliefs and the role they played within their management of chronic low back pain |

Continued

**Table 2** Continued

| Author, year | Country | Data collection | Analytical approach | Order of analysis and professional group/context | Participants | Study focus |
|---|---|---|---|---|---|---|
| Dobbs, 2014[172] | USA | Focus groups | Content analysis | 10. Pain in age care facilities | 28 Nursing assistants | To explore: (1) communication about pain between nursing home residents and nursing assistants; (2) how race and ethnicity influence experiences; and (3) assistants' pain experiences that affect their empathy |
| Eccleston, 1997[151] | UK | Q-analysis | Q-analysis | 2. Mixed HCPs | 11 Mixed HCPs (five anaesthetists, four psychologists, one nurse, one physiotherapist) | To explore how sense is made of the causes of chronic pain |
| Espeland, 2003[173] | Norway | Focus groups | Phenomenology | 9. Guidelines | 13 GPs | To explore: (A) that affect GPs' decisions about ordering X-rays for back pain and (B) barriers to guideline adherence |
| Esquibel, 2014[174] | USA | Interviews | Immersion–crystallisation | 8. Opioid prescription | 21 Family practitioners (10 residents, 6 attending) | To explore the experiences of adults receiving opioid therapy for relief of chronic non-cancer pain and that of their physicians |
| Fontana, 2008[175] | USA | Semistructured interview | Emancipatory research | 8. Opioid prescription | 9 Advanced practice nurses | To explore factors that influence the prescribing practices of advanced practice nurses for patients with chronic non-malignant pain |
| Fox, 2004[176] | Canada | Focus groups | Thematic analysis | 10. Pain in age care facilities | 54 Mixed HCPs (13 healthcare aides, 8 registered practice nurses, 19 registered nurses, 6 physicians, 8 OTs/PTs) | To explore barriers to the management of pain in long-term care institutions |
| Gooberman-Hill, 2011[177] | UK | Semistructured interview | Thematic analysis | 8. Opioid prescription | 27 GPs | To explore GPs' opinions about opioids and decision-making processes when prescribing 'strong' opioids for chronic joint pain |
| Gropelli, 2013[150] | USA | Semistructured interviews | Content analysis | 10. Pain in age care facilities | 16 Nurses (registered and licenced practical nurses) | To explore nurses' perceptions of pain management in older adults in long-term care |
| Hansson, 2001[178] | Sweden | Interviews | Grounded theory | 7. Pain-related work disability | 4 Physicians | To explore life lived with recurrent, spine-related pain and to explore the development from work to disability pension |
| Harting, 2009[179] | The Netherlands | Focus groups | Content analysis | 9. Guidelines | 30 Physiotherapists | To explore the determinants of guideline adherence among physical therapists |
| Hayes, 2010[180] | Canada | Focus groups and interviews | Grounded theory | 5. Fibromyalgia | 32 Physicians (GPs, rheumatologists, psychiatrists, neurologists, anaesthesiologists) | To explore knowledge and attitudinal challenges affecting optimal care in fibromyalgia |
| Hellman, 2015[181] | Sweden | Semistructured interviews | Thematic analysis | 7. Pain-related work disability | 15 Mixed HCPs (four OTs, four PTs, two social workers, three physicians, two psychologists) | To explore health professionals' experience of working with return to work (RTW) in multimodal rehabilitation for people with non-specific back pain |

Continued

**Table 2** Continued

| Author, year | Country | Data collection | Analytical approach | Order of analysis and professional group/context | Participants | Study focus |
|---|---|---|---|---|---|---|
| Hellström, 1998[182] | Sweden | Interviews | Phenomenology | 5. Fibromyalgia | 20 Physicians (10 rheumatologists, 10 GPs) | To explore the clinical experiences of doctors when meeting patients with fibromyalgia |
| Holloway, 2009[183] | Australia | Semistructured interviews | Constant comparison | 10. Pain in age care facilities | 6 Nursing assistants | To explore the experiences of nursing assistants who work with older people in residential aged care facilities (chronic pain example) |
| Holloway, 2009*[184] | Australia | Semistructured interviews | Constant comparison | 10. Pain in age care facilities | 6 Nursing assistants | To explore the experiences of nursing assistants who work with older people in residential aged care facilities (chronic pain example) |
| Howarth, 2012[185] | UK | Interviews and focus groups | Grounded theory | 6. Chronic pain services | 9 Mixed HCPs (consultant nurse, physiotherapist, two consultant psychologists, two pain nurses, three anaesthetists) | To explore person-centred care from the perspectives of people with chronic back pain and the interprofessional teams who care for them |
| Kaasalainen, 2007[186] | Canada | Interviews and 8 focus groups | Grounded theory | 10. Pain in age care facilities | 66 Mixed HCPs (physicians (n=9), registered practical nurses) | To explore the decision-making process of pain management of physicians/nurses and how their attitudes about pain affect their decisions about prescribing among older adults in long-term care |
| Kaasalainen, 2010[187] | Canada | Interviews and focus groups | Thematic analysis | 10. Pain in age care facilities | NK Mixed HCPs (registered nurses, personal support workers, nurse practitioners, physicians, pharmacist, PTs, clinical nurse specialists) | To explore the perceptions of healthcare team members who provide care for residents and nurse managers views regarding the nurse practitioner role in pain management in long-term care |
| Kaasalainen, 2010[188] | Canada | Interviews and focus groups | Case-study analysis | 10. Pain in age care facilities | 53 Mixed HCPs (15 registered nurses, 6 registered practical nurses, 4 physicians, 20 unlicenced care practitioners, 2 pharmacists, 2 PTs, 4 administrators) | To explore barriers to pain management in long-term care and develop an interprofessional approach to improve pain management |
| Kilaru, 2014[189] | USA | Semistructured interview | Grounded theory | 8. Opioid prescription | 61 Emergency physicians | To explore themes regarding emergency physicians' definition, awareness, use and opinions of opioid-prescribing guidelines |
| Krebs, 2014[190] | USA | Semistructured interview | Immersion–crystallisation | 8. Opioid prescription | 14 Primary care physicians | To explore physicians' and patients' perspectives on recommended opioid management practices and to identify potential barriers/facilitators of guideline-concordant opioid management in primary care |
| Kristiansson, 2011[191] | Sweden | Interviews | Narrative analysis | 1. Primary care physicians/gps | 5 GPs | To explore GPs' experience in contact with chronic pain patients and what works and does not work in these consultations |
| Liu, 2014[192] | Hong Kong | Interviews and focus groups | Content analysis | 10. Pain in age care facilities | 49 Nursing assistants | To explore nursing assistants' roles during the process of pain management for residents |

Continued

**Table 2** Continued

| Author, year | Country | Data collection | Analytical approach | Order of analysis and professional group/context | Participants | Study focus |
|---|---|---|---|---|---|---|
| Löckenhoff, 2013[193] | USA | Focus groups | Content analysis | 2. Mixed HCPS | **44 Mixed HCPs** (21 physicians, 23 physical therapists) | To explore how perceptions of chronological time influence the management of chronic non-cancer pain in middle-aged and older patients |
| Lundh, 2004[194] | Sweden | Focus groups | Constant comparison | 1. Primary care physicians/gps | **14 GPs** | To explore what it means to be a GP meeting patients with non-specific muscular pain |
| Macneela, 2010[149] | Ireland | Critical incident interview | Thematic analysis | 1. Primary care physicians/gps | **12 GPs** | To explore how GPs represent chronic low back pain especially in relation to psychosocial care |
| McConigley, 2008[195] | Australia | Interviews and focus groups | Thematic analysis | 10. Pain in age care facilities | **34 Mixed HCPs** (7 GPs, 11 registered nurses, 4 enrolled nurses, 8 allied health professionals, 4 facility managers) | To develop recommendations and a 'toolkit' to facilitate implementation of pain management strategies in Australian Residential Aged Care Facilities |
| McCrorie, 2015[196] | UK | Focus groups | Grounded theory | 8. Opioid prescription | **15 GPs** | To explore the processes that bring about and perpetuate long-term prescribing of opioids for chronic, non-cancer pain |
| Mentes, 2004[197] | USA | Semistructured interviews | Thematic analysis | 10. Pain in age care facilities | **11 Certified nursing assistants** | To evaluate pain information from formal direct caregivers who cared for cognitively impaired residents |
| O'Connor, 2015[198] | USA | Ethnography | Constant comparison | 6. Chronic pain services | **NK Mixed HCPs** (psychiatrist, chiropractor acupuncturist, yoga/massage/ craniosacral/ movement/ massage/ occupational therapists, medical director, health coach, nutritionist) | To explore patterns of communication and decision making among clinicians collaborating in the care of challenging patients with chronic low back pain |
| Øien, 2011[148] | Norway | Interviews, focus groups, observation | Case study | 3. Physiotherapists | **6 Physiotherapists** | To explore communicative patterns about change in demanding physiotherapy treatment situations |
| Oosterhof, 2014[199] | The Netherlands | Interviews and observation | Thematic analysis | 6. Chronic pain services | **10 Mixed HCPs** (three OTs, onr rehabilitation physician, three physiotherapists, two psychologists, one social worker) | To explore which factors are associated with a successful treatment outcome in chronic pain patients and professionals participating in a multidisciplinary rehabilitation programme |
| Parsons, 2012[200] | UK | Semistructured interviews | Framework analysis | 2. Mixed HCPs | **19 Mixed HCPs** (5 osteopaths, 4 chiropractors, 10 physiotherapists) | To explore beliefs about chronic muscular pain and its treatment and how these beliefs influenced care seeking and process of care |
| Patel, 2008[201] | UK | Semistructured interview | Thematic analysis | 1. Primary care physicians/GPs | **18 GPs** (11 South Asian/7 white British) | To explore GPs' experiences of managing patients with chronic pain from a South Asian community |
| Patel, 2009*[202] | UK | Semistructured interview | Thematic analysis | 1. Primary care physicians/gps | **18 GPs** (11 South Asian/7 white British) | To explore the experiences and needs for management of people from a South Asian community who have chronic pain |

Continued

**Table 2** Continued

| Author, year | Country | Data collection | Analytical approach | Order of analysis and professional group/context | Participants | Study focus |
|---|---|---|---|---|---|---|
| Paulson, 1999[203] | Sweden | Interviews | Phenomenology | 5. Fibromyalgia | **21 Mixed HCPs** (17 nurses, 4 physicians) | To explore the experiences of nurses and physicians in their encounter with men with fibromyalgia |
| Poitras, 2011[204] | Canada | Semistructured interviews | Thematic analysis | 9. Guidelines | **9 Occupational therapists** | To identify barriers and facilitators related to the use of low-back pain guidelines as perceived by OTs |
| Ruiz, 2010[205] | USA | Focus groups and interviews | Grounded theory | 9. Older adults | **19 Mixed HCPs** (14 primary care physicians, 5 nurse practitioners) | To explore the attitudes of primary care clinicians towards chronic non-malignant pain management in older people |
| Schulte, 2010[206] | Germany | Semistructured interview | Thematic analysis | 1. Primary care physicians/gps | **10 GPs** | To explore the factors that influence whether referrals from GPs are made, and at what stage, to specialised pain centres |
| Scott-Dempster 2014[207] | UK | Semistructured interviews | IPA | 4. Specialist physiotherapists | **6 Physiotherapists** | To explore physiotherapists' experiences of using activity pacing with people with chronic musculoskeletal pain |
| Seamark, 2013[208] | UK | Interviews and focus groups | Thematic analysis | 8. Opioid prescription | **22 GPs** | To explore factors influencing GPs' prescribing of strong opioid drugs for chronic non-malignant pain |
| Shye, 1998[209] | USA | Focus groups | Immersion–crystallisation | 9. Guidelines | **22 Primary care physicians** | To explore why an intervention to reduce variability in imaging rates for low back pain was ineffective among physicians |
| Siedlecki, 2014[210] | USA | Interviews | Grounded theory | 11. Nursing | **48 Nurses** | To explore nurses' assessment and decision-making behaviours related to the care of patients with chronic pain in the acute care setting |
| Slade, 2012[144] | Australia | Focus groups | Grounded theory | 3. Physiotherapists | **23 Physiotherapists** | To explore how physiotherapists prescribe exercise for people with non-specific chronic low back pain in the absence of definitive or differential diagnoses |
| Sloots, 2009[211] | The Netherlands | Semistructured interviews | Thematic analysis | 6. Chronic pain services | **4 Rehabilitation physicians** | To explore factors lead to tension in the patient–physician interaction in the first consultation by rehabilitation physicians of patients with chronic non-specific low back pain of Turkish and Moroccan origin |
| Sloots, 2010[212] | The Netherlands | Semistructured interviews | Thematic analysis | 6. Chronic pain services | **10 Mixed HCPs** (8 rehabilitation physicians, 1 PT, 1 OT) | To explore which factors led to drop-out in patients of Turkish and Moroccan origin with chronic non-specific low back pain in a rehabilitation programme |
| Spitz, 2011[213] | USA | Focus groups | Thematic analysis | 8. Opioid prescription | **26 Mixed HCPs** (23 physicians, 3 nurse practitioners) | To explore primary care providers' experiences and attitudes towards prescribing opioids as a treatment for chronic pain among older adults |
| Starrels, 2014[214] | USA | Telephone interview | Grounded theory | 8. Opioid prescription | **28 Physicians** | To explore primary care providers' experiences, beliefs and attitudes about using opioid treatment agreements for patients with chronic pain |

Continued

**Table 2** Continued

| Author, year | Country | Data collection | Analytical approach | Order of analysis and professional group/context | Participants | Study focus |
|---|---|---|---|---|---|---|
| Stinson, 2013[215] | Canada | Focus groups | Thematic analysis | 6. Chronic pain services | **17 Mixed HCPs** (6 physicians, 4 registered nurses, 3 physiotherapists, 1 pharmacist, 1 chiropractor, 1 marriage and family therapist, 1 OT) | To explore the information and service needs of young adults with chronic pain to inform the development of a web-based chronic pain self-management programme |
| Thunberg, 2001[147] | Sweden | Interviews | Grounded theory | 6. Chronic pain services | **22 Mixed HCPs** (seven physicians, three psychologist, two physiotherapists, eight nurses, two social workers) | To explore the way HCPs perceive chronic pain |
| Toye, 2015[216] | UK | Focus groups | Grounded theory | 2. Mixed HCPs | **19 Mixed HCPs** (11 GPs, 3 nurses, 3 pharmacists, 1 physiotherapist, 1 psychiatrist) | To explore the impact on HCPs of watching and discussing a short film about patients' experience of chronic MSK pain |
| Tveiten, 2009[217] | Norway | Focus groups | Content analysis | 6. Chronic pain services | **5 mixed HCPs** (medicine, nursing physiotherapy) | To explore the dialogue between the health professionals and the patient at a pain clinic |
| Wainwright, 2006[145] | UK | Interviews | Thematic analysis | 1. Primary care physicians/GPs | **14 GPs** | To explore the dilemma of treating medically explained upper limb disorders |
| Wilson, 2014[143] | UK | Interviews, letters, documents | Ethnography | 9. Guidelines | **NK Mixed HCPs** (involved in the debate) | To explore the meaning of the guideline and the sociopolitical events associated with it |
| Wynne-Jones, 2014[218] | UK | Semistructured interviews | Constant comparison | 7. Pain-related work disability | **17 Mixed HCPs** (11 GP, 6 physiotherapists) | To explore GPs' and physiotherapists' perceptions of sickness certification in patients with musculoskeletal problems |
| Zanini, 2014[219] | Italy | Semistructured interviews | Thematic analysis | 6. Chronic pain services | **17 Physicians** (12 rheumatology, 2 neurology, 1 immunology, 1 psychiatry, 1 'nervous and mental disease') | To explore aspects that are important to address during a consultation to build a partnership with patients with chronic pain |

*Sample reported in two papers.
GPs, general practitioners; HCPs, healthcare professionals; IPA, interpretative phenomenological analysis; LBP, low back pain; MSK, musculoskeletal; NK, not known; OTs, occupational therapists; PT, physiotherapist.

**Table 3** Confidence in review findings: GRADE–CERQual assessment

| Review finding | Adequacy number of concepts | Coherence* number of studies/77 | Methodological limitations n=satisfactory (n=key) | Relevance | Overall assessment of confidence |
|---|---|---|---|---|---|
| Sceptical cultural lens | 43 | 29[147 149 151 153 156 159 161 166–168 170–174 180 182 183 188 190 191 194 196 200 210 213–216] | 29 (0) | 22 direct, 4 indirect, 2 partial, 1 uncertain | Moderate |
| Navigating juxtaposed models of medicine | 77 | 44[144–149 152 153 155–158 162 165 168–170 173 174 182 191 193 194 196 199 204 206 207 209 210 212 215–219] | 42 (2) | 37 direct, 4 indirect, 2 partial, 1 uncertain | High |
| Navigating the geography between patient and HCP | 92 | 36[144–149 152 153 155–158 162 165 168–170 173 174 182 191 193 194 196 199 204 206 207 209 210 212 215–219] | 34 (2) | 29 direct, 3 indirect, 3 partial, 1 uncertain | Moderate |
| The craft of pain management | 60 | 31[143 144 149 150 153–156 158 159 168 170–173 175–177 179 185 188 194 195 199 200 204 205 208 210 213 218] | 29 (2) | 27 direct, 2 indirect, 1 partial, 1 uncertain | Moderate |
| Challenge of dual advocacy | 70 | 36[144 147 149 150 156 159 160 162–164 166–168 170 176 178 181 183–186 188 191 192 195–198 202 204–206 209 210 215 218] | 35 (1) | 26 direct, 4 indirect, 5 partial, 1 uncertain | Moderate |
| Personal cost | 71 | 33[146 148 152 153 155 156 158 159 161 162 165 166 170–172 176 182–184 186 191 192 194 196 197 201 207 210 215–219] | 32 (1) | 28 direct, 4 indirect, 1 partial | Moderate |

*15/371 concepts did not fit conceptual categories.
HCPs, healthcare practitioners.

who may be more likely to adopt a biopsychosocial approach.[146 147 155 157 164 185 198 199 211 212 215 217 219]

## A sceptical cultural lens

This theme describes a culturally entrenched sceptical view of chronic non-malignant pain from which HCPs did not always trust patients' reports of pain. This lack of trust meant that clinical work involved determining whether the pain was *something* or *nothing*. HCPs found themselves making judgements, based on personal factors rather than clinical findings, about whether the pain was *real* or *imagined*. They pondered dissonance between what the patient said and what the HCP could see.

> Sometimes I could have a patient sitting there and saying that they are hurting, 10 out of 10, and they are sitting like you and I.[159] (Bergman, 2013, primary care, GP, USA)

> Some people say 'This is the worst pain I've had in my whole life' without any real sort of physical signs of pain so it's really tough; we have a complex job in assessing that.[188] (Kaasalainen, 2010, aged care facilities, unspecified HCP, Canada)

There was a sense that HCPs were 'on guard'[149] against exploitation from fraudulent claims. For example, the following family practitioners felt concerned about being 'manipulated' or 'exploited' by patients:

> It is not clear to me why he is the way he is… this catastrophic pain and what he is telling himself about it… but there is always a little bit… of concern; am I being manipulated, is this really real?[174] (Esquibel, 2014, opioid prescription, family practitioner, USA)

> Such people… ones whose wishes you cannot fathom – provoke anger and frustration because at some point, you don't always know how to verify their complaints. You feel somewhat exploited. It is a very unpleasant feeling.[170] (Dahan, 2007, guidelines, GP, Israel)

HCPs engaged in a process of categorisation to decipher patients' truth claims. This categorisation hinged on deciphering a multiplicity of dualities that were superimposed on a polarity of '*good*' and '*bad*'. HCP described these dualities as follows: easy/difficult, explained/not explained, local/diffuse pain, adherent/non-adherent, stoical/weak, motivated/unmotivated, accepting/resisting, non-complaining/complaining and deserving/non-deserving. Some recognised that this categorisation was flawed and advocated trust as the basis of the therapeutic relationship. For example, the following HCPs describe how at times they had made the wrong judgement about patients who were truly in pain:

> Sometimes we say 'oh she came in with back pain but I don't think she's really in pain'… but really even if somebody is in pain and distress, [it] doesn't always have to be in how they present themselves… that doesn't mean she is not in pain.[216] (Toye, 2015, mixed HCPs on a pain education course, unspecified HCP, UK)

> I hate to say it… but I used to be one of the people that used to say, 'Oh, well, they are probably just wanting attention.' But I've changed in that matter. People are in pain, and it's not just to get attention.[167] (Clark, 2006, age care facilities, unspecified HCP, USA)

## Navigating juxtaposed models of medicine

This describes the challenge of navigating juxtaposed models of medicine: the biomedical and the biopsychosocial. The biomedical model takes disease to be an objective biomedical category not influenced by psychosocial factors, whereas the biopsychosocial model incorporates psychosocial factors influencing the pain experience.[220] There was a culturally entrenched pull towards the biomedical *siren song* of diagnosis.[144]

> Being able to track something gives me more comfort than going by what you're telling me… because I like to see proof… You [want to]… be convinced that you're treating *something* and that what you're treating is real.[158] (Berg, 2009, opioid prescription, physician, USA)

> I will listen to their story, I will examine them and I always say you have got to exclude the physical first that is your job… we have an obligation to exclude the physical first and not jump into [psychosocial explanations] because it reduces the patient to being an un-necessary complainer and I don't believe that they really are.[145] (Wainwright, 2006, primary care, GP, UK)

The following GP describes how chronic pain can obscure 'real' tangible health problems (such as high cholesterol) with the implication that chronic pain is less real:

> They don't seem to worry about issues that might be real… like his cholesterol is high… there are some other issues that he needs to attend to… his father died when he was fifty two. He's not worried.[159] (Bergman, 2013, primary care, GP, USA)

Some HCPs used a dualistic biopsychosocial model, whereby once *something* biomedical has been excluded, they made an abrupt shift towards psychosocial explanations. Here clinical work shifted away from diagnosis towards persuading patients that psychosocial factors influenced pain. This abrupt shift could threaten the therapeutic relationship, and HCPs described how psychological explanations came with a stigma attached.

> The terminology… psychiatric and psychological… have a stigma attached to them that is not intended… we accept that patients with long term pain will have a psychological component to it but actually labelling it as that.[216] (Toye, 2015, mixed HCPs on a pain education course, unspecified HCP, UK)

HCPs therefore tended to default to physical explanations or used 'bridging' strategies to keep the patient on board. The following HCPs described the importance of approaching psychological explanations in a very careful way:

It is a subtlety and if you present [the explanation for pain] as a completely airy fairy psychological, it is up to you, then they are going to go away dissatisfied, so you have got to lead them in gently.[145] (Wainwright, 2006, primary care, GP, UK)

If you start from the body and if you ask a little carefully how things work when you are physically like that, then it is not threatening, and you can approach things, like, through the body.[146] (Afrell, 2010, pain specialist, physiotherapist, Norway)

Some felt that a diagnosis could help a patient to move forward or give a sense of relief. One HCP described how they might 'feign diagnostic certainty' to achieve this goal.[145] Another HCP voiced ethical concerns about the deception of *feigning* diagnostic certainty.

I think giving it a label that actually has no justification I think is misleading to the patient and I actually feel quite strongly about that.[145] (Wainwright, 2006, primary care, GP, UK)

Not all HCPs used a dualistic biopsychosocial model and did not make this abrupt shift towards psychosocial explanations. Rather, they used an embodied biopsychosocial approach with 'no breaking point where the physical becomes psychological'.[157] There was a sense that pain is multidimensional and that the 'physician gaze'[221] is multifocal. Here clinical work involved understanding person's suffering from the outset of care. The following HCPs working in specialist pain services describe feelings of empathy and understanding:

While we talked… many losses came up and I began myself to think about what all this was about in fact, what is this pain? Where it came out that there was a lot of disappointment, where there was divorce and… yes, it can't be purely physiological.[146] (Afrell, 2010, pain specialist, physiotherapist, Norway)

Once a person's life has fallen apart it's not so much about the pain and the injury anymore. It's about all these other things in their life and it's all these other things that need to be addressed in order to get them better and get them back on track.[164] (Cartmill, 2011, chronic pain services, unspecified HCP, Canada)

Those who used a more embodied psychosocial model recognised that sitting alongside and supporting patients, rather trying to 'fix' them, could be rewarding for both the patient and their HCP.

I think the sort of traditional model of treatment doesn't allow people to express how pain has affected their whole life, it is very homed in to the particular area of the body and trying to fix it, and I just find it

more satisfying to work in a way that acknowledges and discusses the impact.[207] (Scott-Dempster, 2014 pain specialist, physiotherapist, UK)

HCPs also described how time restrictions could encourage HCPs to focus on the physical body and were perceived as a barrier to an embodied approach.

We are limited by the amount of time with the patient. I know this sounds bad, but [talking about pain] opens a can of worms.[210] (Siedlecki, 2014, acute care, nursing, USA)

### Navigating the geography between patient and clinician

This describes the complexity of navigating the geography between patient and HCP. The metaphor of geography is used to portray a sense that the terrain could prove treacherous. The following HCP describes patients feeling dissatisfied by the health encounter:

People feel let down by their doctors… The degree of satisfaction is very low… basically because we don't solve their problem… They go from one to the other, they find a doctor who gives them hope and they go to him.[162] (Briones-Vozmediano, 2013, fibromyalgia, occupational health doctor, Spain)

HCPs therefore made concessions in order to navigate the geography between patient and clinician. For example, they might make choices of doubtful medical utility, such as prescribing pain killers or referring for an investigation, in order to maintain relationships. Concessions were sometimes necessary to balance long-term and short-term gains. The following HCP describes referring a patient for a test in order to show the patient that they are listening to them:

Sometimes patients refuse to believe that their condition cannot be treated… and insistently ask for a series of medical investigations that you, as a doctor, would not perform. In such cases, a medical investigation can work as a therapy because it… shows that you listened to them.[219] (Zanini, 2014, chronic pain services, neurology physician, USA)

HCPs also described the personal challenge that accompanied a need to balance professional expertise and patient empowerment. The following HCPs described how they could find it difficult to stand back and let patients make what they felt was a '*wrong*' decision. The following examples highlight HCPs personal struggle with this challenge:

Trying to allow myself to listen objectively and to… sit with the fact that actually [the patient] might want to do something which is wholly *unsensible*, but allowing that to happen if that truly is what they want.[207] (Scott-Dempster, 2014, pain specialist, physiotherapist, UK)

I recognise that… we are trying to promote learning by giving choice and allowing people to get it wrong… we learn by *doing* not by being told what to do. I get

that, although it is still hard… not to give advice when I see… that the advice can be really helpful.[155] (Barker, 2015, pain specialist, physiotherapist, UK)

If a conflict arose, the 'short-circuit'[217] could be to take control, but there remained a sense that discussion rather than enforcement was more effective in the long term. Empowering patients involved helping them to make decisions for themselves with HCP support.

> [If we think] 'I know that this is the correct answer'… then you do not allow the patient to participate. He then becomes a receiver. But if you share your knowledge… then you offer the patient an opportunity to think and decide by himself.[217] (Tveiten, 2009, chronic pain services, unspecified HCP, Norway)

> Patients have to embrace our suggestion because they are convinced that it is the right one and not because we want them to choose a particular option. If you propose something that is inconsistent with their experience or knowledge, there is a risk that they will not listen to you.[219] (Zanini, 2014, chronic pain services, rheumatology physician, USA)

## The challenge of dual advocacy

This theme describes the HCP as being simultaneously an advocate of the patient and an advocate of the healthcare system. While representing the patients' interests, at the same time, HCPs represented the healthcare system and made important decisions as representatives of that system. This resonates with the challenge of making decisions based on what is best for the individual patient alongside utilitarian decisions for the greatest number. At times, this sense of dual advocacy could create an uncomfortable feeling that healthcare colleagues were not working on the same side as each other or the patient. At times, it could feel like the experience was spiralling of control ('a ship without a rudder').

> It all ends up on our doorstep. It is not only we who face the system – we are mediators of sorts between the patient and the system. Not only must we work with the patient against the system, but with the system as well.[170] (Dahan, 2007, guidelines, GP, Israel)

> As soon as someone gets sort of uncomfortable they will shift to a different prescriber and they will push them along a certain course… and I honestly think it's like a ship without a rudder and it's just going round and round in circles.[196] (McCrorie, 2015, opioid prescription, GP, UK)

HCPs also described how it could prove difficult to access specialist pain services and that it could feel like there was a mismatch between what the HCP expected and what they received.

> There is a really big access issue with the pain clinics right now… while I can refer them, their likelihood of getting an appointment, even with strong advocacy from me, is very low… Often I find that they are not accomplishing any more than I was and [patients] are often sent back to me with them essentially saying, 'we did our best.' It's very frustrating, because if they were easy… they wouldn't have been referred.[156] (Barry, 2010, opioid prescription, physician, USA)

HCPs recognised the benefits of a healthcare system where the cogs worked smoothly: the benefits of reciprocity and collaboration, being confident in the capabilities of colleagues and reciprocal respect.

> We get a lot of mileage about slapping each other on the back a little bit. And increasing other members of the team's confidence by respecting other members of the team, their profile is improved.[185] (Howarth, 2012, chronic pain services, physician, UK)

> If the team sort of echoes the same message and provides richness in terms of their different perspective on it… then I think there's less confusion for the poor clients and they're able to follow through on a unified evidence-based recommendation.[164] (Cartmill, 2011, chronic pain services, unspecified HCP, Canada)

## Personal costs

This theme describes the emotional costs of treating patients with chronic pain. First, the biomedical model could create a sense of professional failure for not being able to *fix*; 'how did we fail them?'.[210] This sense of failure could be demoralising and undermine HCPs sense of professionalism.

> You become a doctor not to tell people I can't do anything, I can't find anything, you have this perception of yourself as well that you're going to sort it out and if you can't sort it out, it's frustrating. What's the point of you being there?[201] (Patel, 2008, primary care, GP, UK)

> It's awful, and I think it's demoralizing when you leave people in pain. That's just so disrespectful. I mean you're supposed to be a doctor, you're supposed to relieve pain and suffering, and you ignore the pain.[156] (Barry, 2010, opioid prescription, physician, USA)

However, an embodied biopsychosocial approach that hinged on recognising human losses could incur a deep sense of personal loss. HCPs described how they had to manage the tension between proximity and distance.

> We forget how much chronic pain affects the patient. They lose their jobs, they have emotional stress and depression and the depression itself is a big loss of productivity to the patient but also to the entire family and to the community.[210] (Siedlecki, 2014, acute care, nurse, USA)

> Trying to listen to the person… sort of empathise… [but] almost protected professionally… trying to see where that person was coming from but not letting it become too personal… I've used the phrase detached

empathy.[216] (Toye, 2015, mixed HCPs on a pain education course, unspecified HCP, UK)

### The craft of pain management

This describes clinical work as an experience-based competence or 'craft'[171] gained from experience rather than didactic education. At times, HCPs felt underskilled in chronic pain management.

> The problem is, we don't know how to treat pain. And so everybody is telling me I'm not treating pain well, but nobody is helping me figure out how to treat the pain.[213] (Spitz, 2011, opioid prescription, physician, USA)

Personal experience or maturity, amount of experience treating patients with chronic pain and learning from colleagues underpinned craft knowledge.

> One becomes more stable as a person [with age], and does not really have the same demands and does not believe that one can do everything, that one is able to solve everything… Young doctors can have in them, that they believe that they will solve everything.[153] (Asbring, 2003, fibromyalgia, physician, USA)

> New grads can't learn all of this, they need a certain number of years, you can't teach them all of this.[144] (Slade, 2012, physiotherapists, Australia)

Although some HCPs felt that clinical guidelines could support a more patient-centred approach,[179 204] there was a stronger sense that they constrained craft knowledge. HCPs therefore used guidelines pragmatically within the remit of their own knowledge.

> Treatment has to be tailored to patient's needs and prescriptive guidelines promoting 'one size fits all' is not acceptable.[143] (Wilson, 2014, guidelines, unspecified HCP, UK)

> If you work according to the guidelines, you are constrained in your performance… what would be left of your independence, your own competence, your own practical experience… Am I to conclude then that my training was useless… I'm free to take or leave these things, to look at whether they suit my own ideas of how to approach my patients.[179] (Harting, 2009, guidelines, physiotherapist, Netherlands)

### Line of argument

The final phase of meta-ethnographic analysis is to develop a model or *line of argument* that is abstracted from, *but more than the sum of,* the themes (figure 2).[11] Through discussion with each other, and the advisory group, the reviewers developed a model that made sense of the final themes. The model is underpinned by a series of tensions that can help us to understand and reflect on the experience of treating patients with chronic non-malignant pain: (A) between a dualistic biomedical model and an embodied psychosocial model; (B) between professional distance and proximity; (C) between professional expertise and patient

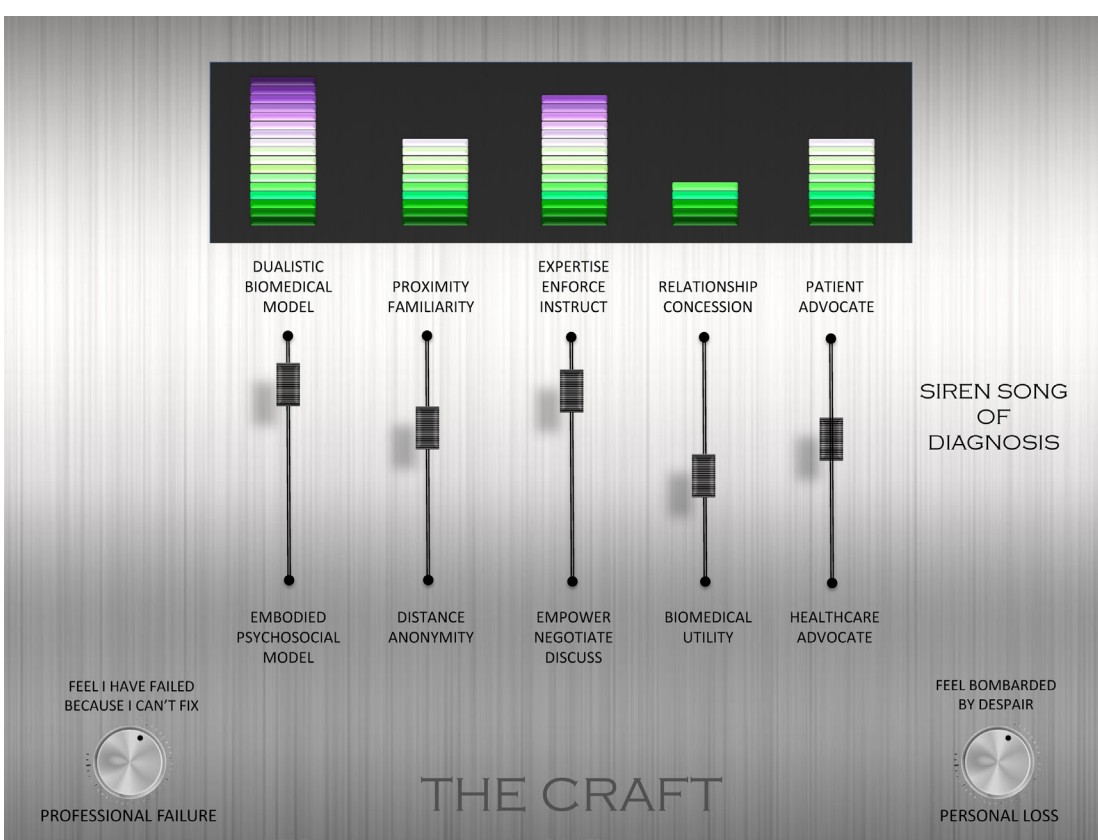

**Figure 2** Line of argument.

empowerment; (D) between a need to make concessions in order to maintain relationships and known biomedical utility; and (E) between patient and healthcare system advocacy. We conceptualise these tensions, on a mixing console,[i] as underpinning the craft of chronic non-malignant pain management. The poles are neither inherently *good* nor *bad*; just as bass and treble are neither inherently good nor bad. It is the correct mix within a context that contributes to the quality of music. The levels indicated in figure 2 are an example and do not indicate any sense of 'correct' balance. Different HCPs may adjust the balance differently for each individual and context. Our console also incorporates the *pitch* or level of loss, both professional and personal, that can contribute to the harmony or dissonance of a therapeutic encounter. The siren song of diagnosis, reflecting the cultural pull of the biomedical model, is also shown as a factor that can have an important impact on the balance between poles.

## DISCUSSION

Our innovation is to present the first internationally relevant QES of HCPs' experiences of treating chronic non-malignant pain. Already we know that, from the patient perspective, this experience can be adversarial.[10] Patients with chronic pain struggle to affirm their sense of self, and their present and future appears unpredictable; they search for a credible explanation for their pain; they do not always feel heard, believed or valued by HCPs; they struggle to prove themselves in the face of scepticism. We present our line of argument as a mixing console that can help HCPs to understand, think about and modify their experience of treating patients with chronic pain. For example, an HCP could consider: am I making a sudden shift to psychosocial explanations when I can find nothing biomedical or am I considering psychosocial factors alongside medical investigations?; do I understand this patient's experience or am I too distant?; have I discussed and negotiated the various options or am I trying to instruct and enforce?; am I considering medical utility or am I making a concession (and is this concession for my benefit or my patient's benefit)?; am I effectively balancing my role as dual advocate? Beyond these dualities, our model encourages HCPs to consider the personal impact of treating patients with chronic non-malignant pain. How often do you find yourself wondering whether you have failed as a professional? (professional loss); are you feeling *bombarded by despair*? (personal loss). If the answer is yes to either of these, what measures are there in place to tackle this?

Culture has been described as the 'inherited lens' through which individuals understand the world and learn how to live in it.[222] Both patients and their HCPs are embedded in a wider culture where biomedical explanations have the power to bestow credibility. The studies included explore the experience of both diagnosing and treating pain and demonstrate that these are inextricably linked. Our findings highlight the cultural scepticism that underpins the *siren song of diagnosis*, where HCPs and patients can be driven by the need for a diagnosis. This may help us to understand why patients with chronic pain often experience a strong sense of not being believed. They also demonstrate how the biopsychosocial model can hide a continuing dualism, where HCPs prioritise biomedical findings and make an abrupt switch to psychosocial explanations when no diagnosis is found. This abrupt shift may explain patients' feeling of lost credibility. A more embodied non-dualistic biopsychosocial approach at the outset would help HCPs to support patients with chronic pain. Our findings also demonstrate the complexity of navigating the geography between patients and HCPs. In this borderland, HCPs sometimes make concessions that are not evidence-based in order to maintain effective relationships. These concessions have policy and practice implications, for example, in the context of recent USA[223] and UK[224] guidelines on opioid prescription for chronic non-malignant pain, it might help to explain why an increasing number of HCPs are prescribing opioids despite very limited evidence for long-term opioid therapy for chronic pain outside of end-of-life care.[225] Our findings also have educational implications: for example, navigating relationships requires skills to finely balance the tensions inherent in the model while managing potential personal and professional losses. HCPs included in this review did not discuss their own personal life context that intuitively might contribute to a sense of loss. This might indicate that there were topics that were not explored in the initial interview studies and further research might explore the impact of this on HCPs' resilience to challenges of treating people with chronic pain and other conditions. HCPs described experience of treating chronic non-malignant pain that was not boundaried to a particular body system but was a summative experience cutting across conditions. Further research might focus on specific diagnosis (such as neuropathic, visceral, pelvic or phantom pain and arthritis) in order to explore potential similarities and difference in HCP experiences of treating these conditions.

Although we used the GRADE-CERQual approach, there is currently no agreed way to determine confidence in QES findings. It would be useful for future studies to consider the following issues: first, although GRADE-CERQual considers methodological limitations as having an impact on confidence in reviews, there is limited agreement about what a good qualitative study is.[29 31] Indeed, a significant number of QES reviewers choose not to appraise studies.[24] Although quality appraisal might highlight methodological flaws, it does not necessarily help us to appraise the usefulness of findings for the purposes of QES. It could be argued that good studies are excluded if our primary concern is methodology rather than conceptual insight.[24 31] It would be useful for future studies to address how reviewers can be more discerning about the value of particular studies and the influence on analytical decision. This issue will become more important as the number of primary qualitative

---

[i] Idea for image of a mixing console from Cathy Jenkins, OUH NHS Foundation Trust, Oxford bmj open supplementary appendix 1 1march17.pdf

research studies grows. Although our reviewers agreed about which studies were 'key', 'fatally flawed' or 'irrelevant',[21] the majority of studies were appraised as 'satisfactory'. As only five studies were appraised as 'key', this status did not influence data analysis. Second, GRADE-CERQual considers adequacy (weight) and coherence (consistency) of data as important factors that can contribute to confidence in a review finding. However, do these necessarily equate to validity and how do we know *what* is adequate? The issue of determining adequacy resonates with the unresolved question 'how many qualitative interviews is enough?'.[226] We chose to rate our confidence in a finding as high when a theme was supported by a least half of the studies (n≥39). However, although you could argue that weight and consistency[32] of findings contribute to the persuasiveness of a finding, it is important to consider that a unique idea can exert a significant pull. It is thus important not to ignore unique or inconsistent findings. We have found that confidence in QES findings can grow when you incorporate a large number of studies. However, QES reviewers can be caught between a rock and a hard place as they face criticisms for undertaking reviews that are 'too small' (and thus anecdotal) or 'too large' (not in-depth). Another potential criticism of a QES that includes a large number of studies is that it is possible to lose sight of the nuances of the primary studies. We found that using NVivo qualitative analysis software allowed us to keep track of our analytical decisions while being able to continually refer back to the primary studies. This helped us to ensure that our findings remained grounded in these primary studies.

Findings from QES in health aim to provide ideas that can help to improve the experience of healthcare. We have presented a novel line of argument that helps us to understand, think about and modify our experience of diagnosing and treating patients with chronic non-malignant pain. Our line of argument may be transferable to other patient groups or situations. We conceptualise dualities, on a mixing console, as a useful way to frame the patient–clinician relationship. It would be useful for HCPs to consider their individual mix and contemplate a *re-mix* if necessary in order to successfully support people with chronic pain. Now we have a body of qualitative knowledge exploring patients' experiences of chronic pain[9] and HCPs' experiences; the next challenge in practice is to bring these two bodies of knowledge together and look at how HCPs and patients can work together in managing pain.

**Contributors** All authors made a substantial contribution to the design, acquisition, analysis and interpretation of data. FT drafted the first, subsequent and final versions. KS and KLB revised all versions for important intellectual content and approved the final version. All authors agree to be accountable for the accuracy and integrity of the work.

**Funding** This study was funded by the National Institute for Health Research Health Services and Delivery Research Programme (14/198/07).

**Disclaimer** The views and opinions expressed therein are those of the authors and do not necessarily reflect those of the HS&DR Programme, NIHR, NHS or the Department of Health. KLB and FT authored 2 of the 77 studies included in this review.

**Competing interests** None declared.

**Provenance and peer review** Not commissioned; externally peer reviewed.

**Data sharing statement** The original protocol, full monograph and supporting material can be found at: http://www.nets.nihr.ac.uk/projectsOld/hsdr/1419807.

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
