## [Reviewer comments · BMJ Open]

ARTICLE DETAILS

TITLE (PROVISIONAL)	A meta-ethnography to understand healthcare professionals' experience of treating adults with chronic non-malignant pain
AUTHORS	Toye, Francine; Seers, Kate; Barker, Karen L.

VERSION 1 – REVIEW

REVIEWER	Bridget Laging La Trobe University; Australian Catholic University, Melbourne Australia.
REVIEW RETURNED	20-Jul-2017

GENERAL COMMENTS	This paper provides valuable insight into the experiences of healthcare professionals who care for with people with chronic pain. The following recommendations have been made to enhance the quality of this paper. 1. Research question and study objective A definition of chronic, non-malignant pain is needed. Some background on the context of chronic pain diagnosis and treatment issues is also needed. The following sentence needs to be clarified “Each year over five million people develop chronic pain”. Is this worldwide? The relevance of the following sentence to your topic is not clear. “For example, medicine taking [4], diabetes [5] antidepressants [6], osteoporosis [7]chronic musculoskeletal pain [2, 8] and chronic pelvic pain [9].” 2. Abstract: Remove abbreviations from abstract (HCPs) Consider replacement of “conceptual synthesis of qualitative research” with meta-synthesis. Who are HCPs? It would be useful to clarify who this group consists of in the abstract? Clarify “We rated confidence in review findings as moderate (5 findings) or high (1 finding)” It is not clear what is meant by ‘internationally relevant’. Further clarification is needed to support this statement. Conclusion: The phrase ‘findings highlight’ is repeated. Consider condensing these two sentences. Also this sentence is an example of anthropomorphism. Consider rephrasing this to acknowledge the role of the researchers who analysed this data.
---

This is the first internationally relevant synthesis of healthcare professionals' experiences of treating people with chronic non-malignant pain. We have presented a conceptual model that helps us to understand, think about and modify our experience of treating patients with chronic pain. Our findings highlight cultural scepticism that might explain why patients with chronic pain feel they are not believed. Findings highlight a potential dualism in the bio-psychosocial model and the complexity of navigating therapeutic relationships. Our conceptual model may be transferable to other patient groups or situations.

4. Methods:

p.4 28 What is meta-ethnography? A clearer definition is required here.

In the abstract it is stated that articles were collected from the inception to present time. This is not elaborated on in the methods section. It is not clear why this time period has been selected. Why not a later date? Also, in the findings, it would be useful to identify when were most studies were published.

p.4 34-38 It is stated that there are seven stages to meta-ethnography, but only six are listed.

p.5 26 In the introduction to the method the authors state that meta-ethnography goes beyond description to develop conceptual understandings. The authors need to describe in this section how this was undertaken. Further description of how constant comparative analysis was undertaken to develop initial concepts into themes is also needed.

p.5 p.35 A reference is made to Figure 2. This figure needs a heading and the flow chart needs to be adjusted to match the PRISMA chart.

p.5, 42. Further clarification on the difference between the following reasons for exclusion is needed. What is the difference between the reasons for exclusion for the following: and two as irrelevant [133, 134] out of scope [96-122].?

p.5 44, Authors state that they appraised five studies as 'key papers' What criteria was used to identify 'key papers'? How did this influence analysis of the data?

p.5, 49-50 "doctors, nurses and allied health professionals in various contexts and geographical locations". Further in text detail would be useful here. Who was most readily represented in the samples? How many doctors, nurses, allied health professionals took part in each study? Was there sufficient data from the primary studies to provide this information?

A definition of allied health professionals is also needed to identify which disciplines were included.

A brief overview of the main contexts and geographical locations would also be useful in text.

p. 5 50 “Table 1 shows the author and year of publication, geographic context, number of participants, data collection method, analytic approach and sequence of analysis”. This does not match the description on page 29 “Table 1: Geographic context, sample size, data collection and analytic approach, order of analysis and professional group/topic”

Table One requires significant review.

The table provides an inadequate summary of the included studies. Consider the following headings: 1. Author/year/country 2. Study focus 3. Contributing participants (including numbers of doctors, nurses and allied health professionals) 4. Design: data collection and analysis 5. Substantial findings.

p.5 54-60 Excellent identification and great scope for future research.

p. 6, 8 A reference to a link for a supplementary file needs to be provided for the reader to view how the 371 concepts were developed into 42 conceptual categories

5. Ethics: Were the primary studies screened to check that they had received ethical approval? Were they excluded if ethical approval had not been obtained?

6. yes.

7. N/a

8. Yes

9. Findings – do the findings address the research question?

The aim of this paper is to present “A meta-ethnography to understand healthcare professionals’ experience of treating adults with chronic non-malignant pain”

The title of Figure 3, which depicts the conceptual model developed from the meta-synthesis is entitled ‘Siren Song of Diagnosis’.

This does not clearly link with the initial aim, and there needs to be some clarification regarding whether the focus of this paper is on identifying and diagnosing chronic pain or the treatment. My feeling is that it is the former.

10. Are the findings presented clearly?

p.6, 28 “This describes a cultural lens that provides a skeptical view of chronic non-malignant pain” it is not clear what a cultural lens is. It is also not obvious to the reader what is meant by a skeptical view. Further description of these concepts would enhance understanding of this theme. Also, the term “non-clinical judgement” is ambiguous and requires clarification. The introduction to this theme could be enhanced with reference to the lack of trust that HCPs have in what patients say regarding their pain.

Whilst quotes are supported by references to the studies, the findings would be enhanced if there was also identification of the participant discipline from the original study.

p. 7, 1. What is ‘boundary work’? A definition is needed here.

p. 7, 2. Which studies did the following concepts arise from?: This boundary work hinged on a multiplicity of dualities superposed on a polarity of ‘good’ and ‘bad’ (easy/difficult; explained/not explained, local/diffuse pain; adherent/non-adherent; stoical/weak; motivated; unmotivated; accepting/resisting; non-complaining/complaining; deserving/non-deserving).

p. 7, 27 A brief description of the two models: the biomedical and the bio-psychosocial, is needed in the introduction of this theme to provide a reference for the reader as to how the two models differ.

p.8, 59 “Some felt that a diagnosis could help a patient to move forward, or give a sense of relief. Some even ‘feign[ed] diagnostic certainty’ to achieve this goal [137].

The fact that they tell you that you have a problem that’s not just to do with your nerves and that there’s something wrong physically . . . Just that gives you a certain sense of relief[166].”

Clarification of participants is needed here: “Some x felt...”. Some [insert people who are being referred to here] even ‘feign[ed]....”.

Again, the provision of details of the participant from the original study would enhance the findings dramatically.

p.9, 33 remove “/”

p. 10, 11. Authors state “This theme describes the HCP as simultaneously advocate of the patient and the healthcare system”. Further clarification regarding which aspects of the healthcare system are being referred to here are needed. What is meant by dual advocacy? How did the findings reflecting advocating for the health system? It would appear rather, that the findings regarding the health system are related to the presence or absence of collegial support.

p.11, 1 “This theme describes the HCP as simultaneously advocate of the patient and the healthcare system.” Authors need to review grammatical formation of this sentence.

p.13, 8-14.

I am not a psychologist I don’t know whether it is fair to expect me to do all of that and I don’t know if anyone is expecting me to. . . Someone bringing out a lot about their past or perhaps a very complex situation . . . we don’t want to say the wrong thing and it be to someone’s detriment . . . you don’t want to open this can of worms[174].

This quote is poorly linked to the theme. Consider removing.

p. 14, 17-25

The poles are neither inherently good nor bad; just as bass and treble are neither inherently good nor bad. It is the correct mix within a context that contributes to the quality of music. Our console also incorporates the pitch or level of loss, both professional and personal, that can contribute to the harmony or dissonance of a therapeutic encounter.

This is a really beautiful analogy. Further description of Figure 3 is needed. How did the authors arrive at the conclusion of where to place the knobs on the five scales?

11. Discussion and justification

Again, clarification regarding whether this paper is about diagnosis or treatment is needed.

p.15, 18 Do any studies exist that have explored or implemented a bio-psychosocial approach exist in the literature? Support from these would be useful here.

A recommendation is made by the authors that "It might be useful for clinical educators to consider overlaps in training need between palliative care and chronic pain management". The link with palliative care is not clear here, as this has not been introduced in the paper until now. What educational preparation exists at present for HCPs regarding chronic pain? Some discussion around this would be helpful.

p.15, 37-38 HCPs did not discuss "their own personal life context which is also likely, at times, to contribute and make HPCs less resilient to personal or professional losses. This identifies a limitation in the studies included. Do other studies support this statement that personal life influences HCP experiences in management of chronic pain? Or, is this a gap in the literature? Clarification needed here.

12. Limitations:

p. 15, 42-55 This detail belongs in the section discussing the critique on the quality of the studies. "We need to consider that there is currently no agreed way of making an assessment of confidence for qualitative synthesis. We utilised the GRADE-CERQual to rate confidence as high when a theme was supported by a least half of the studies (n≥39)".

"There are issues to be considered in our use of GRADE-CERQual: First, we know that there is limited agreement about what a good qualitative study is [22, 24]. Secondly, although weight (adequacy) and consistency (coherence) [25] of data can contribute to the gravity of a finding, it is important to consider that gravity has a qualitative component; a single idea can exert a strong gravitational pull. The Tale of the Emperor's New Clothes highlights the validity of a small voice in the crowd".

Further clarification of what you are trying to say here is needed. Are you saying that although greater validity can be associated with consistency of findings across studies is important, the value of stand alone findings is also valuable? If this is so, is a weakness of the study the exclusion of findings that are different to those that were not consistently identified?

Another weakness of the review that needs to be considered is the limited description of participants. It is unclear if the perspectives represented are shared by doctors, nurses and allied health professionals. It is also unclear if each professional group was represented in the findings, or whether the findings are more representative of one professional group.

REVIEWER	Padraig MacNeela National University of Ireland, Galway Ireland
REVIEW RETURNED	08-Aug-2017

GENERAL COMMENTS	This study is a meta ethnography of a very large number of studies - 77 studies. Phrasing is at times casual, for instance, the statement 'healthcare as adversarial' p.4 – may have adversarial aspects but to describe healthcare as a whole in this way is not a suitable way to open. The study refers to health care professionals – HCPs, although no discrimination is made between different professional groups and settings within health care, which I think is a critical point given the breadth covered in the review. The strategy is referred to as 'conceptual synthesis', and then the authors move on – I bring up this example as in many instances the authors need to further unpack what is meant. There is a very brief background given on past research, which I believe needs more development – what is chronic pain, what are the categories of HCPs involved, what is their role in terms of provision of care (medication, psychosocial, informal support; community / institutional / specialised; acute / chronic). More discussion and summary is required in text of the range of number of studies included in this method. What proportion were in primary care, what proportion had mixed health care professional background as opposed to one discipline ... Psychinfo – incorrect spelling The phrase 'Previous experience' p.4 – can you unpack this, do you mean these authors in particular? What categories did the studies fall into with respect to the illness / form of chronic pain? Reference made to the authors having 'worked with a project advisory group' – what did this consist of? What's the basis for the conceptual analysis, what was the emerging model or conceptualization that guided the researchers in filtering and configuring the 371 elements to 42 larger constructs and then to 6? The process is not sufficiently documented. Later at the end of the findings we hear the analogy of a 'mixing desk' – is this intended to grow from the analysis or is something that just occurs as a good idea to the authors? The analysis phase of building the metaethnography is presented in the findings, should be in method. The findings refer to 'Cultural lens' – is this referring to culture outside of healthcare or healthcare culture? The discrimination between 'something' and 'nothing' seems false – assessing more for degrees of pain or for degrees of impact of pain. HCP listed as HPC on Page 6, also Page 7, 9 careless unless I am missing something? Page 7: 'superposed'
--

Why is the boundary work 'moral'

Is the strategy on page 8 actually 'concealment' or a professional strategy of getting the patient on board

When saying 'some felt' on page 8 I am wondering what the difference is here from a thematic analysis – you said you wanted to portray a conceptual model, but here I don't see that followed through if you are simply documenting the proportion of studies that refer to a particular point, particularly if phrased in terms of 'some felt' it comes across like a thematic account of a particular group of participants.

Page 9 – 'the geography between patient and HPC' – phrasing seems awkward, what is a 'borderland'

Reflecting on the overall presentation of results, I would prefer to see fewer quotes but more done with them – basic rule of thumb used in this paper is to present two quotes for each assertion, but the reader is left to make the connection between the concept introduced in the preceding paragraph e.g. the point introduced about borderland and the quote that is supposed to support this. Would be more effective to have fewer quotes but then to see the interpretive work done to show what is 'borderland' about the quote in question. In other cases there may be differences between quotes being used to illustrate diversity in the concept, but as it is there are two quotes applied to each assertion without reference to the content really at any time. If you look at page 13 it is mainly comprised of quotes, is this study one where you develop a concept or make short assertions and provide implicit evidence

Homogeneity / heterogeneity is a key issue here – given the breadth of the investigation I would expect some diversity, culturally, professionally between different disciplines, treatment contexts, patient groups – but we get a unitary construct – e.g., professionals felt patients might make 'wrong' decisions

Missing syntax on first line of theme introduced on page 11

Page 13 – 'patient mileage' – another example of a phrase that I would rephrase or if deemed valuable to unpack sufficiently for meaning.

Personally I'm not too impressed with the 'mixing console' analogy, I didn't get a lead up during the themes that this was heading to the idea that the clinicians are formulating 'music', as in particular most of the commentary was negative – talking about challenges, difficulties etc. – mostly dissonant rather than 'music'. Plus, seeing the elements of care as elements of music implies a deterministic position, this is neither 'good nor bad'. Overall what is the implication for the patients' rights and interests stemming from your model, and should this implication be accommodated within your model, or are you suggesting it is value free?

Such a view is possible but I don't see it argued in the methodology. I see a vision of the HCP as the mixer and manager and creator of the music to be a power laden view, this is continued in the discussion where the 'I' is the voice of the clinician who seems to be assumed to be the person making all the decisions about what mix is required.

While this might reflect the ideology inherent in the quotes, I think it is an important part of the concept to reflect this implicit position of power as it resonates with your opening statement about patients' view of being in an adversarial position.

The concept of duality is good but does it imply there is duality within clinicians or between clinicians when considered as a whole group – is there any of this that can be traced back to context, setting, professional discipline etc.?

Confused by reference to Emperor's new clothes on page 15

Can we hear in the discussion about specific implications or suggestions for research, about reconciling to some degree with patient literature, and limitations / critical reflection on this piece of research – e.g. use of 77 studies in a meta synthesis must present certain challenges.

The use of HCP as a term ultimately rings a bit hollow for me in the discussion with the reference to prescribing of opioids which positions the main meaning of HCP as a medical role i.e. the doctor as opposed to other professions.

Overall I feel that the paper has merits in its efforts to assimilate literature on this critically important topic. However I wonder if the stage presented by a set of 77 studies is being oversimplified or to what degree the meta ethnography is the most appropriate strategy – I would like to have this explained and argued more strongly. I find the presentation of the findings does include conceptual organisation yet what is being done in the presentation of the individual points and use of quotes strikes me as simplistic, very little attention to nuances of language use in the quotes. I am also reiterating the issue about the diversity of contexts, disciplines, etc. and how this is not at all apparent when the authors present their synthesis. I find some aspects of the discussion disheartening – the centrality of the practitioner as the determiner of the care process and reference to 'I' too closely aligning authors with the practitioners, and little critical questioning of HCP role and power. Finally I don't feel the mixing desk analogy follows naturally from the findings and feels a bit bolted on.

VERSION 1 – AUTHOR RESPONSE

Reviewer: 1 Bridget Laging

We are pleased that you feel our paper provides valuable insight into the experiences of healthcare professionals who care for with people with chronic pain.

1. Research question and study objective

A definition of chronic, non-malignant pain is needed. Some background on the context of chronic pain diagnosis and treatment issues is also needed.

We have added the following in response to both reviewers' comments:

Chronic pain is defined as pain which persists beyond the three months expected time of healing [1]. In 2009, an estimated five million people in the UK develop chronic pain [2], and a recent systematic review suggests that this may underestimate the problem [3]. Around 20% of adults in Europe have chronic pain[4] and in the USA more than 25 million adults (11%) experience chronic pain[5]. Chronic pain is challenging because it persists beyond healing time and is not easy to explain or treat. A range of clinical staff are involved in caring for people with chronic pain and in the UK there is a wide range in the provision of specialist care [6]. Not all patients with chronic pain have access to specialist services, and a national UK audit in 2012 indicated that only 40% of pain clinics met the minimum stand of having a psychologist, physiotherapist and physician[6]. The audit suggests that as many as 20% of patients with chronic pain visit accident and emergency departments even after visiting their general practitioner (GP), and as many as 66% visit a clinician three times within a six month period.

2. The following sentence needs to be clarified "Each year over five million people develop chronic pain". Is this worldwide?

We have now clarified this (see response 1).

3. The relevance of the following sentence to your topic is not clear. "For example, medicine taking [4], diabetes [5] antidepressants [6], osteoporosis [7]chronic musculoskeletal pain [2, 8] and chronic pelvic pain [9]."

The relevance of this is to give readers some examples of where meta-ethnography has been used in health research. We have rewritten to clarify this:

We aimed to conduct a QES using the methods of meta-ethnography [7] Meta-ethnography is widely used and has provided insight into healthcare experiences such as; medicine taking [8], diabetes [9] antidepressants [10], osteoporosis [11]chronic musculoskeletal pain [12, 13] and chronic pelvic pain [14].

4. Abstract:

Remove abbreviations from abstract (HCPs)

Thanks. We have removed these.

5. Consider replacement of "conceptual synthesis of qualitative research" with meta-synthesis.

We have replaced throughout with qualitative evidence synthesis (QES) which is now commonly used.

6. Who are HCPs? It would be useful to clarify who this group consists of in the abstract?

This is an important point raised by both reviewers. The study included a wide range of HCPs and we have now included much more detail on this. We have added to the study table details of numbers of different HCPs (where available) and also indicated the source of each narrative exemplar chosen to illustrate the themes.

In the abstract we have added:

The experiences of over 1551 international healthcare professionals including doctors, nurses and other healthcare professionals.

7. Clarify “We rated confidence in review findings as moderate (5 findings) or high (1 finding)”

To simplify, we have reworded this in the abstract and explain in full in the body text:

We rated confidence in review findings as moderate to high.

We rated our confidence in the review finding as high when it was supported by more than half of the studies ($n \geq 39$). However, there is currently no agreed way of making an assessment of confidence for QES.

8. It is not clear what is meant by ‘internationally relevant’. Further clarification is needed to support this statement.

We have removed ‘internationally relevant’ from abstract, and indicate the countries from which the synthesis was drawn in the main body text (see below). We also indicate country of origin in each narrative exemplar.

We included 77 published studies reporting the experiences of over 1551 HCPs from USA (20 studies), UK (18 studies), Canada (10 studies), Sweden (10 studies), Netherlands (4 studies), Norway (4 studies), Australia (4 studies), France (1 study), Germany (1 study), Hong Kong (1 study), Ireland (1 study), Israel (1 study), Italy (1 study) and Spain (1 study) (table 1).

9. Conclusion: The phrase ‘findings highlight’ is repeated. Consider condensing these two sentences.

We have removed duplication and reworded:

This is the first Qualitative Evidence Synthesis of healthcare professionals’ experiences of treating people with chronic non-malignant pain. We have presented a model that we developed to help healthcare professionals to understand, think about and modify experiences of treating patients with chronic pain. Our findings highlight scepticism about chronic pain that might explain why patients feel they are not believed. Findings also indicate a dualism in the bio-psychosocial model and the complexity of navigating therapeutic relationships. Our model may be transferable to other patient groups or situations.

10. Also this sentence is an example of anthropomorphism. Consider rephrasing this to acknowledge the role of the researchers who analysed this data.

We have rewritten to clarify (see response 9).

Methods:

11. p.4 28 What is meta-ethnography? A clearer definition is required here.

We have developed the methods section for clarity:

Meta-ethnography is a method developed by Noblit and Hare which aims to synthesise qualitative research findings into a whole that is greater than the sum of its original parts[7].

We used the methods of Meta-ethnography developed, refined and reported by Toye and colleagues [12, 15]. There are various methods for synthesising qualitative research [16-20]. An important distinction is between (a) those that describe findings and (b) those, like meta-ethnography, that develop conceptual understandings through a process of constant comparison and abstraction [7]. There are seven stages to meta-ethnography: getting started, deciding what is relevant, reading the studies, determining how studies are related, translating studies into each other, synthesising translations and expressing the synthesis [15].

12. In the abstract it is stated that articles were collected from the inception to present time. This is not elaborated on in the methods section. It is not clear why this time period has been selected. Why not a later date? Also, in the findings, it would be useful to identify when were most studies were published.

We have addressed this in the methods section and have now indicated when most studies were published:

Six studies were published before 2000; 37 were published between 2000 and 2010; 34 were published from 2011 onwards.

13. p.4 34-38 It is stated that there are seven stages to meta-ethnography, but only six are listed.

Thanks, we missed an important step – reading:

“There are seven stages to meta-ethnography: getting started, deciding what is relevant, reading the studies, determining how studies are related, translating studies into each other, synthesising translations and expressing the synthesis”

14. p.5 26 In the introduction to the method the authors state that meta-ethnography goes beyond description to develop conceptual understandings. The authors need to describe in this section how this was undertaken. Further description of how constant comparative analysis was undertaken to develop initial concepts into themes is also needed.

We have developed the methods section in response to both reviewers:

We planned to develop a line of argument synthesis, which involves ‘making a whole into something more than the parts alone imply’ [7] (page 28). This is achieved by comparing concepts and developing ‘a grounded theory that puts the similarities and differences between studies into interpretive order’ [7] (page 64). Analysis in large QES involves a process of: identifying concepts from qualitative studies; abstracting these concepts into conceptual categories; further abstracting categories into themes; and finally developing a line of argument that makes sense of the themes. We read studies in batches of topic or professional grouping. We did not use an index paper to orientate the synthesis [21], as we felt that this choice can have a dramatic impact on the interpretation [22]. Two reviewers read each paper to identify, describe and list concepts. If they agreed that there was no clear concept then it was excluded. Through constantly comparing and discussing concepts three reviewers abstracted concepts into conceptual categories, using NVivo 11 software for qualitative analysis to keep track of our analytical decisions [23]. NVivo is particularly useful for collaborative analysis as it allows the team to keep a record and compare interpretations. Once we had agreed and defined conceptual categories, these were printed onto cards and sent to our advisory group to read and sort into thematic groups. This group consisted of patients, allied health professionals, nursing professionals, doctors and health service managers. Then, during an advisory meeting, the reviewers worked alongside the advisory group to finalise the themes that would be included in the line of argument [7]. In this way, we were able to challenge our own interpretations. The final analytic stage, ‘synthesising translations’ involved the three reviewers working together alongside the advisory group to craft the final themes into a ‘line of argument’. Some reviewers do not present a line of argument as

part of their QES findings. Frost and colleagues indicate that there has been a move away from interpretation and theory development in QES towards aggregative forms of synthesis [24].

15. p.5 p.35 A reference is made to Figure 2. This figure needs a heading and the flow chart needs to be adjusted to match the PRISMA chart.

We have added a heading and revised language to be in line with PRISMA.

16. p.5, 42. Further clarification on the difference between the following reasons for exclusion is needed. What is the difference between the reasons for exclusion for the following: and two as irrelevant [133, 134] out of scope [96-122]?

Thank you, both regard out of scope studies. We have amended the numbers in the text and on flowchart:

p.5 44, Authors state that they appraised five studies as 'key papers' What criteria was used to identify 'key papers'? How did this influence analysis of the data?

This is a global assessment of key based on work by Mary-Dixon Woods. You raise an important unresolved methodological point here and we have expanded this in the discussion.

It would be useful for future studies to address how reviewers can be more discerning about the value of particular studies and the influence on analytical decision. Although our reviewers agreed about which studies were 'key', 'fatally flawed' or 'irrelevant', [18] as only five studies were appraised as 'key' this status did not influence data analysis. This issue will become more important as the number of primary qualitative research studies grows.

17. p.5, 49-50 "doctors, nurses and allied health professionals in various contexts and geographical locations". Further in text detail would be useful here. Who was most readily represented in the samples? How many doctors, nurses, allied health professionals took part in each study? Was there sufficient data from the primary studies to provide this information?

This is a very important point. It was not possible to identify exact numbers from the primary papers. We have included much more detail to table 1 as suggested, contextualised narrative exemplars and added:

Table 1 shows the author, year of publication, country, sample size, data collection, analytic approach, and professional group/context, participants and study focus. The studies included explored the experience of: General Practitioners (10 studies); mixed HCPs in diverse contexts (4 studies); physiotherapists (3 studies); physiotherapists with a speciality in chronic pain (3 studies); mixed HCPs in Fibromyalgia (5 studies); mixed HCPs in chronic pain services (11 studies); mixed HCPs in pain management related to employment (5 studies); mixed HCPs prescribing opioids to patients with chronic pain (12 studies); mixed HCPs utilising guidelines for chronic pain (6 studies); mixed HCPs working with older adults with chronic pain (3 studies); mixed HCPs working in long term care facilities (13 studies); nurses (2 studies).

18. A definition of allied health professionals is also needed to identify which disciplines were included.

We have removed allied and replaced this with 'other health professionals' and also included participant information in the revised table.

19. A brief overview of the main contexts and geographical locations would also be useful in text.

We agree that this is important contextual information and have added the geographical locations (see question 8 above) and also the study focus in (revised table 1).

20. p. 5 50 “Table 1 shows the author and year of publication, geographic context, number of participants, data collection method, analytic approach and sequence of analysis”. This does not match the description on page 29 “Table 1: Geographic context, sample size, data collection and analytic approach, order of analysis and professional group/topic”

We have changed both to:

Table 1 shows the author and year of publication, country, data collection method, analytic approach, professional group and context, participants and study focus

21. Table One requires significant review.

The table provides an inadequate summary of the included studies. Consider the following headings:

1. Author/year/country 2. Study focus 3. Contributing participants (including numbers of doctors, nurses and allied health professionals) 4. Design: data collection and analysis 5. Substantial findings.

We agree that this table gives only selected information and have significantly reviewed the table to include, author and year of publication, country, data collection method, analytic approach, professional group and context, participants and study focus. We have not included the substantial findings from each study as we feel that, for a QES of this size, this would make this table beyond a reasonable size for this publication. However, a monograph giving the information that you suggest is being published by the NIHR who were the funders of this study. We provide the readers a link for this information.

22. p.5 54-60 Excellent identification and great scope for future research.

Thank you.

23. p. 6, 8 A reference to a link for a supplementary file needs to be provided for the reader to view how the 371 concepts were developed into 42 conceptual categories

A monograph giving the information that you suggest is being published by the funders (NIHR). We provide the link to this:

A full report giving further details of analytical decisions is being published by the NIHR Journals library <https://www.journalslibrary.nihr.ac.uk/programmes/hsdr/1419807/#/>

24. Ethics: Were the primary studies screened to check that they had received ethical approval? Were they excluded if ethical approval had not been obtained?

You raise an important issue. Although some journals are not explicit about ethical approval requirements, we did screen potential studies for ethical standards (CASP question 7: Have ethical issues been taken into consideration

We have added (method):

Although some journals are not explicit about ethical approval requirements, we screened potential studies for ethical standards (CASP question 7: Have ethical issues been taken into consideration?).

And (findings):

We agreed that ethical issues had been satisfactorily considered in the study design of all 77 studies and none were excluded on ethical grounds.

25. Findings – do the findings address the research question?

The aim of this paper is to present “A meta-ethnography to understand healthcare professionals' experience of treating adults with chronic non-malignant pain”

The title of Figure 3, which depicts the conceptual model developed from the meta-synthesis is entitled 'Siren Song of Diagnosis'. This does not clearly link with the initial aim, and there needs to be some clarification regarding whether the focus of this paper is on identifying and diagnosing chronic pain or the treatment. My feeling is that it is the former.

We have now clarified this as follows:

[In findings] We aimed to explore healthcare professionals' experience of treating chronic non-malignant pain. We found that studies explored the experiences of diagnosing and treating chronic pain, and that these experiences were inextricably linked.

[In discussion] Culture has been described as the 'inherited lens' through which individuals understand the world and learn how to live in it [25]. Both patients and their HCPs are embedded in a wider culture where biomedical explanations have the power to bestow credibility. Studies explore the experience of both diagnosing and treating pain and demonstrate that these are inextricably linked. Our findings highlight the cultural scepticism that underpins the siren song of diagnosis, where HCPs and patients can be driven by the need for a diagnosis. This may help us to understand why patients with chronic pain often experience a strong sense of not being believed. They also demonstrate how the bio-psychosocial model can hide a continuing dualism, where HCPs prioritise biomedical findings and make an abrupt switch to psychosocial explanations when no diagnosis is found.

'Siren song of diagnosis' is not the title of this figure so we have amended to clarify this and provided a title (Figure 3 - line of argument). Rather, the siren song of diagnosis is an influencing factor that can have an important impact on the balance between poles.

Are the findings presented clearly?

26. p.6, 28 “This describes a cultural lens that provides a sceptical view of chronic non-malignant pain” it is not clear what a cultural lens is. It is also not obvious to the reader what is meant by a sceptical view. Further description of these concepts would enhance understanding of this theme. We have defined the use of cultural lens in the discussion section and we have also reworded in line

with your suggestion below to emphasis the lack of trust:

This theme describes a culturally entrenched sceptical view of chronic non-malignant pain from which HCPs did not always trust patients' reports of pain. This lack of trust meant that clinical work involved determining whether the pain was something or nothing. HCPs found themselves making judgements, based on personal factors rather than clinical findings, about whether the pain was real or imagined. They pondered dissonance between what the patient said and what the HCP could see.

27. Also, the term “non-clinical judgement” is ambiguous and requires clarification.

We have reworded as suggested (response 27)

28. The introduction to this theme could be enhanced with reference to the lack of trust that HCPs have in what patients say regarding their pain.

We have reworded along these lines (response 27)

29. Whilst quotes are supported by references to the studies, the findings would be enhanced if there was also identification of the participant discipline from the original study.

We have now added detail on professional group in revised table

30. p. 7, 1. What is 'boundary work'? A definition is needed here.

Thank you. Boundary work refers to the process of categorisation. We have rewritten to clarify this concept:

HCPs engaged in a process of categorisation to decipher patients' truth claims. This categorisation hinged upon deciphering a multiplicity of dualities which were superimposed on a polarity of 'good' and 'bad'. HCP described these dualities as follows: easy/difficult; explained/not explained, local/diffuse pain; adherent/non-adherent; stoical/weak; motivated/ unmotivated; accepting/resisting; non-complaining/complaining; deserving/non-deserving. Some recognised that this categorisation was flawed and advocated trust as the basis of the therapeutic relationship.

31. p. 7, 2. Which studies did the following concepts arise from?: This boundary work hinged on a multiplicity of dualities superposed on a polarity of 'good' and 'bad' (easy/difficult; explained/not explained, local/diffuse pain; adherent/non-adherent; stoical/weak; motivated; unmotivated; accepting/resisting; non-complaining/complaining; deserving/non-deserving).

The studies supporting each concept are indicated in table 2. This is integral to the 'sceptical cultural lens' which arose from the 29 studies shown in table 2 [26-57]. We now clarify this as follows: "The studies supporting each theme are shown in table 2."

32. p. 7, 27 A brief description of the two models: the biomedical and the bio-psychosocial, is needed in the introduction of this theme to provide a reference for the reader as to how the two models differ.

We have added:

"The biomedical model takes disease to be an objective biomedical category not influenced by psychosocial factors, whereas the bio-psychosocial model incorporates psychosocial factors influencing the pain experience"

33. p.8, 59 "Some felt that a diagnosis could help a patient to move forward, or give a sense of relief. Some even 'feign[ed] diagnostic certainty' to achieve this goal [137].

The fact that they tell you that you have a problem that's not just to do with your nerves and that there's something wrong physically . . . Just that gives you a certain sense of relief[166]."

Clarification of participants is needed here: "Some x felt...". Some [insert people who are being referred to here] even 'feign[ed]...".

Again, the provision of details of the participant from the original study would enhance the findings dramatically.

We agree and have now done this in the revised table

34. p.9, 33 remove "/"

We have removed this.

35. p. 10, 11. Authors state "This theme describes the HCP as simultaneously advocate of the patient and the healthcare system". Further clarification regarding which aspects of the healthcare system are being referred to here are needed. What is meant by dual advocacy? How did the findings reflecting advocating for the health system? It would appear rather, that the findings regarding the health system are related to the presence or absence of collegial support.

We have rewritten this theme for clarity:

This theme describes the HCP as being simultaneously an advocate of the patient and an advocate of the healthcare system. Whilst representing the patients' interests, at the same time, HCPs represented the healthcare system and made important decisions as representatives of that system. This resonates with the challenge of making decisions based on what is best for the individual patient alongside utilitarian decisions for the greatest number. At times this sense of dual advocacy could create an uncomfortable feeling that healthcare colleagues were not working on the same side as each other or the patient. At times, it could feel like the experience was spiralling of control ('a ship without a rudder')

36. p.11, 1 "This theme describes the HCP as simultaneously advocate of the patient and the healthcare system." Authors need to review grammatical formation of this sentence.

We have rewritten this theme for clarity (see response 36)

37. p.13, 8-14.

I am not a psychologist I don't know whether it is fair to expect me to do all of that and I don't know if anyone is expecting me to. . . Someone bringing out a lot about their past or perhaps a very complex situation . . . we don't want to say the wrong thing and it be to someone's detriment . . . you don't want to open this can of worms[174].

This quote is poorly linked to the theme. Consider removing.

Thanks. We have removed it

38. p. 14, 17-25

The poles are neither inherently good nor bad; just as bass and treble are neither inherently good nor bad. It is the correct mix within a context that contributes to the quality of music. Our console also incorporates the pitch or level of loss, both professional and personal, that can contribute to the harmony or dissonance of a therapeutic encounter.

This is a really beautiful analogy. Further description of Figure 3 is needed. How did the authors arrive at the conclusion of where to place the knobs on the five scales?

Thank you, we are very pleased that you think so.

We have clarified this as follows:

The levels indicated in figure 3 are an example and do not indicate any sense of 'correct' balance. Different HCPs may adjust the balance differently for each individual and context.

39. Discussion and justification

Again, clarification regarding whether this paper is about diagnosis or treatment is needed.

This is an important point and we have clarified and developed it in the discussion (see response 26)

40. p.15, 18 Do any studies exist that have explored or implemented a bio-psychosocial approach exist in the literature? Support from these would be useful here.

Yes, themes are drawn from a range of approach. The revised table 1 gives details of professionals and context. We have added to the discussion:

The themes are drawn from a wide range of HCPs, including those specialising in chronic pain management who may be more likely to adopt a biopsychosocial approach [52, 54, 57-69]

41. A recommendation is made by the authors that “It might be useful for clinical educators to consider overlaps in training need between palliative care and chronic pain management”. The link with palliative care is not clear here, as this has not been introduced in the paper until now. What educational preparation exists at present for HCPs regarding chronic pain? Some discussion around this would be helpful.

We have removed the reference to palliative care as unsubstantiated. We have added: A survey of undergraduate pain curricula for healthcare professionals in the UK [70] indicates that although these curricula are available, pain education is highly variable and ‘woefully inadequate given the prevalence and burden of pain’ [70] (page 78).

42. p.15, 37-38 HCPs did not discuss “their own personal life context which is also likely, at times, to contribute and make HCPs less resilient to personal or professional losses. This identifies a limitation in the studies included. Do other studies support this statement that personal life influences HCP experiences in management of chronic pain? Or, is this a gap in the literature? Clarification needed here.

We have clarified as follows:

HCPs included in this review did not discuss their own personal life context which intuitively might contribute to a sense of loss. This might indicate that there were topics that were not explored in the initial interview studies and further research might explore the impact of this on HCPs’ resilience to challenges of treating people with chronic pain and other conditions.

Limitations:

43. p. 15, 42-55 This detail belongs in the section discussing the critique on the quality of the studies. “We need to consider that there is currently no agreed way of making an assessment of confidence for qualitative synthesis. We utilised the GRADE-CERQual to rate confidence as high when a theme was supported by a least half of the studies ($n \geq 39$)”.

“There are issues to be considered in our use of GRADE-CERQual: First, we know that there is limited agreement about what a good qualitative study is [22, 24]. Secondly, although weight (adequacy) and consistency (coherence) [25] of data can contribute to the gravity of a finding, it is important to consider that gravity has a qualitative component; a single idea can exert a strong gravitational pull. The Tale of the Emperor’s New Clothes highlights the validity of a small voice in the crowd”. Further clarification of what you are trying to say here is needed. Are you saying that although greater validity can be associated with consistency of findings across studies is important, the value of stand-alone findings is also valuable? If this is so, is a weakness of the study the exclusion of findings that are different to those that were not consistently identified?

This is a very important methodological point and we have developed it more fully in the discussion as follows:

Although we utilised the GRADE CERQual approach, there is currently no agreed way to determine confidence in QES findings. It would be useful for future studies to consider the following issues: First, although GRADE CERQual considers methodological limitations as having an impact on confidence in reviews, there is limited agreement about what a good qualitative study is [71, 72]. Indeed, a significant number of QES reviewers choose not to appraise studies [73]. Although quality appraisal might highlight methodological flaws, it does not necessarily help us to appraise the usefulness of findings for the purposes of QES. It could be argued that good studies are excluded if our primary concern is methodology rather than conceptual insight [71, 73]. It would be useful for future studies to address how reviewers can be more discerning about the value of particular studies and the influence on analytical decision.

This issue will become more important as the number of primary qualitative research studies grows. Although our reviewers agreed about which studies were 'key', 'fatally flawed' or 'irrelevant'[18] the majority of studies were appraised as 'satisfactory'. As only five studies were appraised as 'key' this status did not influence data analysis. Secondly, GRADE CERQual considers adequacy (weight) and coherence (consistency) of data as important factors that can contribute to confidence in a review finding. However do these necessarily equate to validity and how do we know what is adequate? The issue of determining adequacy resonates with the unresolved question 'how many qualitative interviews is enough?'[74]. We chose to rate our confidence in a finding as high when a theme was supported by a least half of the studies ($n \geq 39$). However, although you could argue that weight and consistency [75] of findings contribute to the persuasiveness of a finding, it is important to consider that a unique idea can exert a significant pull. It is thus important not to ignore unique or inconsistent findings. We have found that confidence in QES findings can grow when you incorporate a large number of studies. However, QES reviewers can be caught between a rock and a hard place as they face criticisms for undertaking reviews that are 'too small' (and thus anecdotal) or 'too large' (not in-depth). Another potential criticism of a QES that includes a large number of studies is that it is possible to lose sight of the nuances of the primary studies. We found that using Nvivo qualitative analysis software allowed us to keep track of our analytical decisions whilst being able to continually refer back to the primary studies. This ensured that we kept sight of the primary studies.

44. Another weakness of the review that needs to be considered is the limited description of participants. It is unclear if the perspectives represented are shared by doctors, nurses and allied health professionals. It is also unclear if each professional group was represented in the findings, or whether the findings are more representative of one professional group.

We have now added this detail to the revised table and exemplary quotes.

Reviewer: 2 Pdraig MacNeela

1. This study is a meta-ethnography of a very large number of studies - 77 studies.

Phrasing is at times casual, for instance, the statement 'healthcare as adversarial' p.4 – may have adversarial aspects but to describe healthcare as a whole in this way is not a suitable way to open.

We have rewritten the opening to substantiate this finding:

A recent synthesis of 11 qualitative evidence syntheses (QES) has highlighted the personal challenge of living with chronic non-malignant pain and the loss of personal credibility that is integral to this experience [76]. Findings from a QES of 77 qualitative studies exploring patients experience of living with chronic non-malignant pain also demonstrate that patients can experience healthcare as an adversarial battle [12]. Understanding this experience from the perspective of healthcare professionals (HCPs) will help us to unpick this experience and thus contribute to improvements in care provision.

2. The study refers to health care professionals – HCPs, although no discrimination is made between different professional groups and settings within health care, which I think is a critical point given the breadth covered in the review.

Thank you. This is an important point raised by both reviewers. The study included a wide range of HCPs and we have now included more detail on this. We have added to the study table 1 details of numbers of different HCPs (where available in primary studies) and also indicated the source of each narrative exemplar chosen to illustrate the themes.

3. The strategy is referred to as 'conceptual synthesis', and then the authors move on – I bring up this example as in many instances the authors need to further unpack what is meant.

Thank you. We have now developed the description of our methods to unpack what this means. For clarity, we now refer 'line of argument' throughout rather than conceptual model, which is more in keeping with Noblit and Hare's original text.

4. There is a very brief background given on past research, which I believe needs more development – what is chronic pain, what are the categories of HCPs involved, what is their role in terms of provision of care (medication, psychosocial, informal support; community / institutional / specialised; acute / chronic).

We now provide fuller background:

Chronic pain is defined as pain which persists beyond the three months expected time of healing [1]. In 2009, an estimated five million people in the UK develop chronic pain [2], and a recent systematic review suggests that this may underestimate the problem [3]. Around 20% of adults in Europe have chronic pain[4] and in the USA more than 25 million adults (11%) experience chronic pain[5]. Chronic pain is challenging because it persists beyond

healing time and is not easy to explain or treat. A range of clinical staff are involved in caring for people with chronic pain and in the UK there is a wide range in the provision of specialist care [6]. Not all patients with chronic pain have access to specialist services, and a national UK audit in 2012 indicated that only 40% of pain clinics met the minimum stand of having a psychologist, physiotherapist and physician[6]. The audit suggests that as many as 20% of patients with chronic pain visit accident and emergency departments even after visiting their general practitioner (GP), and as many as 66% visit a clinician three times within a six month period.

Chronic pain is challenging because it persists beyond healing time and is not easy to explain or treat. A range of clinical staff are involved in caring for people with chronic pain and in the UK there is a wide range in the provision of specialist care [6]. Not all patients with chronic pain have access to specialist services, and a national UK audit in 2012 indicated that only 40% of pain clinics met the minimum stand of having a psychologist, physiotherapist and physician[6]. The audit suggests that as many as 20% of patients with chronic pain visit accident and emergency departments even after visiting their general practitioner (GP), and as many as 66% visit a clinician three times within a six month period. A survey of undergraduate pain curricula for healthcare professionals in the UK [70] indicates that although these curricula are available, pain education is highly variable and 'woefully inadequate given the prevalence and burden of pain' [70] (page 78).

5. More discussion and summary is required in text of the range of number of studies included in this method. What proportions were in primary care, what proportion had mixed health care professional background as opposed to one discipline?

This is an important point. The study included a wide range of HCPs and we have now included more detail on this. We have added to the study table (1) details of numbers of different HCPs (where available in primary studies) and also indicated the source of each narrative exemplar chosen to illustrate the themes.

6. Psychinfo – incorrect spelling

Thank you. This has been changed

7. The phrase 'Previous experience' p.4 – can you unpack this, do you mean these authors in particular?

Thank you. This raises an important issue about how wide to cast the search net for QES. We have clarified this as follows:

In their original text, Noblit and Hare do not advocate an exhaustive search [7] and the number of studies included in meta-ethnographies ranges [17, 19, 73]. Unlike quantitative syntheses, qualitative syntheses do not aim to summarise the entire body of available knowledge or make statistical inference. We searched five electronic bibliographic databases (Medline, Embase, Cinahl, PsycINFO, Amed) using terms adapted from the InterTASC Information Specialists' Sub-Group (ISSG) Search Filter Resources [77-80]. We used subject headings and free text terms for qualitative research, combined with subject heading and free text terms for pain (figure 1). We did not include citation checks, hand searching, grey literature or PhDs as in our previous QES 95% of the included studies were identified in the first three databases searched [15]. We included studies that explored the experience of all clinical healthcare staff involved in the care of patients with chronic pain.

8. What categories did the studies fall into with respect to the illness / form of chronic pain?

You make an important point. We have added the study focus into table 1 to allow readers to assess context and transferability of findings. We have also added as a discussion point:

HCPs described experience of treating chronic non-malignant pain that was not boundaried to a particular body system, but was a summative experience that cuts across conditions. Further research might focus on specific diagnosis (such as neuropathic, visceral, pelvic or phantom pain, arthritis) in order to explore potential similarities and difference in HCP experiences of treating these conditions.

9. Reference made to the authors having 'worked with a project advisory group' – what did this consist of?

We have now clarified their role in analysis in the methods section (see response 10 below)

10. What's the basis for the conceptual analysis, what was the emerging model or conceptualization that guided the researchers in filtering and configuring the 371 elements to 42 larger constructs and then to 6? The process is not sufficiently documented. Later at the end of the findings we hear the analogy of a 'mixing desk' – is this intended to grow from the analysis or is something that just occurs as a good idea to the authors?

We have expanded the methods section in response to your helpful comments:

We planned to develop a line of argument synthesis, which involves 'making a whole into something more than the parts alone imply' [7] (page 28). Analysis in large QES involves a process of: identifying concepts from qualitative studies; abstracting these concepts into conceptual categories; further abstracting categories into themes; and finally developing a line of argument that makes sense of the themes. We read studies in batches of topic or professional grouping. We did not use an index paper to orientate the synthesis [21], as we felt that this choice can have a dramatic impact on the interpretation [22]. Two reviewers read each paper to identify, describe and list concepts. If they agreed that there was no clear concept then it was excluded. Through constantly comparing and discussing concepts three reviewers abstracted concepts into conceptual categories, using NVivo 11 software for qualitative analysis to keep track of our analytical decisions [23]. NVivo is particularly useful for collaborative analysis as it allows the team to keep a record and compare interpretations. Once we had agreed and defined conceptual categories, these were printed onto cards and sent to our advisory group to read and sort into thematic groups.

This group consisted of patients, allied health professionals, nursing professionals, doctors and managers. Then, during advisory meeting, the reviewers worked alongside the advisory group to finalise the themes that would be included in the line of argument [7]. In this way, we were able to challenge our own interpretations. Some reviewers do not present a line of argument as part of their QES findings. Frost and colleagues indicate that there has been a move away from interpretation and theory development in QES towards aggregative forms of synthesis [24]. The final analytic stage, 'synthesising translations' involved the three reviewers working together alongside the advisory group to craft the final themes into a 'line of argument' to build up a picture of the whole, grounded in the themes.

11. The analysis phase of building the meta-ethnography is presented in the findings should be in method.

We have revised and removed methods from findings as follows:

Two reviewers identified 371 concepts from the 77 studies included. They organised the 371 concepts into 42 conceptual categories and then into six themes: Fifteen out of 371 concepts did not fit our analysis (appendix 1). There were several topics with insufficient weight to develop robust themes: ethnicity [35, 65, 81-83]; gender [84] and older people [29, 30, 85-91]. These may indicate useful areas of further research. Experience specific to opioid prescribing is reported elsewhere. A short film presenting the key themes is available on YouTube [<https://www.youtube.com/watch?v=477yTJPg10o>]. A full report giving further details of analytical decisions is being published by the NIHR Journals library <https://www.journalslibrary.nihr.ac.uk/programmes/hsdr/1419807/#/>.

The six final themes were: (1) a sceptical cultural lens; (2) navigating juxtaposed models of medicine; (3) navigating the geography between patient and clinician; (4) the challenge of dual advocacy; (5) personal cost; (6) the craft of pain management.

12. The findings refer to 'Cultural lens' – is this referring to culture outside of healthcare or healthcare culture?

We have rewritten to clarify cultural lens in response to both reviewers:

This theme describes a culturally entrenched sceptical view of chronic non-malignant pain from which HCPs did not always trust patients' reports of pain. This lack of trust meant that clinical work involved determining whether the pain was something or nothing. HCPs found themselves making judgements, based on personal factors rather than clinical findings, about whether the pain was real or imagined. They pondered dissonance between what the patient said and what the HCP could see.

We also consider this again in the discussion:

Culture has been described as the 'inherited lens' through which individuals understand the world and learn how to live in it [25]. Both patients and their HCPs are embedded in a wider culture where biomedical explanations have the power to bestow credibility.

13. The discrimination between 'something' and 'nothing' seems false – assessing more for degrees of pain or for degrees of impact of pain.

We did not mean to imply that this should be part of assessment, which should certainly include degrees of pain and impact of pain. We have reworded to clarify this:

This theme describes a culturally entrenched sceptical view of chronic non-malignant pain from which HCPs did not always trust patients' reports of pain. This lack of trust meant that clinical work involved determining whether the pain was something or nothing. HCPs found themselves making judgements, based on personal factors rather than clinical findings, about whether the pain was real or imagined. They pondered dissonance between what the patient said and what the HCP could see.

14. HCP listed as HPC on Page 6, also Page 7, 9

We have corrected these typos.

15. Page 7: 'superposed'

We have changed this to superimposed

16. Why is the boundary work 'moral'

It is moral because it incorporates a categorisation based on non-clinical factors of vice or virtue. We have clarified this:

HCPs engaged in a process of categorisation to decipher patients' truth claims. This categorisation hinged upon deciphering a multiplicity of dualities which were superimposed on a polarity of 'good' and 'bad'. HCP described these dualities as follows: easy/difficult; explained/not explained, local/diffuse pain; adherent/non-adherent; stoical/weak; motivated/unmotivated; accepting/resisting; non-complaining/complaining; deserving/non-deserving. Some recognised that this categorisation was flawed and advocated trust as the basis of the therapeutic relationship.

17. Is the strategy on page 8 actually 'concealment' or a professional strategy of getting the patient on board

We have changed this to 'get the patient on board'.

18. When saying 'some felt' on page 8 I am wondering what the difference is here from a thematic analysis – you said you wanted to portray a conceptual model, but here I don't see that followed through if you are simply documenting the proportion of studies that refer to a particular point, particularly if phrased in terms of 'some felt' it comes across like a theme

We feel that using the terms: 'some . . . others . . .' is not incompatible with conceptual abstraction. It is important to keep hold of the diversity of views that can underpin a developing idea. We do not attempt to document the proportion of HCPs who hold to a particular point as this is not the aim of qualitative analysis. We indicate which themes are supported by which studies and which provide concepts that do not fit our analysis. Our line of argument follows our conceptual abstraction through to completion.

19. Page 9 – 'the geography between patient and HPC' – phrasing seems awkward, what is a 'borderland'

We have revised for consistency to make borderland 'geography between patient and clinician'.

Reference to geography is important as it portrays the theme developed with our PPI members to indicate terrain that could be difficult at times to traverse. We have added this to clarify:

This describes the complexity of navigating the geography between patient and HCP. The metaphor of geography helps to portray a sense that the terrain could prove treacherous.

20. Reflecting on the overall presentation of results, I would prefer to see fewer quotes but more done with them – basic rule of thumb used in this paper is to present two quotes for each assertion, but the reader is left to make the connection between the concept introduced in the preceding paragraph e.g. the point introduced about borderland and the quote that is supposed to support this. Would be more effective to have fewer quotes but then to see the interpretive work done to show what is 'borderland' about the quote in question.

In other cases there may be differences between quotes being used to illustrate diversity in the concept, but as it is there are two quotes applied to each assertion without reference to the content really at any time. If you look at page 13 it is mainly comprised of quotes, is this study one where you develop a concept or make short assertions and provide implicit evidence

We have considered your point and rewritten the findings using fewer narrative exemplars that are more clearly woven in with the description of the concepts.

21. Homogeneity / heterogeneity is a key issue here – given the breadth of the investigation I would expect some diversity, culturally, professionally between different disciplines, treatment contexts, patient groups – but we get a unitary construct – e.g., professionals felt patients might make ‘wrong’ decisions

Table 2 indicates the studies from which each theme was drawn. In the interests of transparency, we also indicate themes that did not fit our analysis (appendix 1), which is not standard for QES. As suggested by both reviewers, we have substantially amended our table 1 to include details of participants and focus of each review to allow readers to determine transferability.

22. Missing syntax on first line of theme introduced on page 11

Have added ‘an advocate’

23. Page 13 – ‘patient mileage’ – another example of a phrase that I would rephrase or if deemed valuable to unpack sufficiently for meaning.

We have reworded as suggested to:

‘Amount of experience treating patients which chronic pain’

24. Personally I’m not too impressed with the ‘mixing console’ analogy, I didn’t get a lead up during the themes that this was heading to the idea that the clinicians are formulating ‘music’, as in particular most of the commentary was negative – talking about challenges, difficulties etc. – mostly dissonant rather than ‘music’. Plus, seeing the elements of care as elements of music implies a deterministic position, this is neither ‘good nor bad’. Overall what is the implication for the patients’ rights and interests stemming from your model, and should this implication be accommodated within your model, or are you suggesting it is value free? Such a view is possible but I don’t see it argued in the methodology. I see a vision of the HCP as the mixer and manager and creator of the music to be a power laden view, this is continued in the discussion where the ‘I’ is the voice of the clinician who seems to be assumed to be the person making all the decisions about what mix is required. While this might reflect the ideology inherent in the quotes, I think it is an important part of the concept to reflect this implicit position of power as it resonates with your opening statement about patients’ view of being in an adversarial position. The concept of duality is good but does it imply there is duality within clinicians or between clinicians when considered as a whole group – is there any of this that can be traced back to context, setting, professional discipline etc.?

Your point about power is an interesting one. Our analogy of a mixing console aims to encourage HCPs to contemplate their individual and unique mix between the various poles and to adjust this mix appropriately for each patient. The console does not represent a HCP exerting power and directing the space between patient and clinician, more it provides a way for them to reflect on the balance of these factors that influence their experiences of caring for people with chronic pain. We worked closely with PPI members to ensure that patient voice contributed to the line of argument. The mixing console analogy (reviewer 1 describes as a beautiful analogy) has been used in teaching where it has been an accessible and useful learning tool.

We recognise that lines of argument are not fixed or finite, but rather provide food for thought. We have added this point to our limitations section. We have explicitly stated in the discussion that a particular mix is neither good nor bad.

25. Confused by reference to Emperor's new clothes on page 15

We have removed this

26. Can we hear in the discussion about specific implications or suggestions for research, about reconciling to some degree with patient literature, and limitations / critical reflection on this piece of research – e.g. use of 77 studies in a meta synthesis must present certain challenges.

We have developed the discussion to include challenge and limitations of QES, including: We have found that confidence in QES findings can grow when you incorporate a large number of studies. However, QES reviewers can be caught between a rock and a hard place as they face criticisms for undertaking reviews that are 'too small' (and thus anecdotal) or 'too large' (not in-depth). Another potential criticism of a QES that includes a large number of studies is that it is possible to lose sight of the nuances of the primary studies. We found that using Nvivo qualitative analysis software allowed us to keep track of our analytical decisions whilst being able to continually refer back to the primary studies. This ensured that our findings remained grounded in these primary studies. We also discuss particular implications of quality assessment and confidence in evidence synthesis: Although we utilised the GRADE CERQual approach, there is currently no agreed way to determine confidence in QES findings. It would be useful for future studies to consider the following issues: First, although GRADE CERQual considers methodological limitations as having an impact on confidence in reviews, there is limited agreement about what a good qualitative study is [71, 72]. Indeed, a significant number of QES reviewers choose not to appraise studies [73]. Although quality appraisal might highlight methodological flaws, it does not necessarily help us to appraise the usefulness of findings for the purposes of QES. It could be argued that good studies are excluded if our primary concern is methodology rather than conceptual insight [71, 73]. It would be useful for future studies to address how reviewers can be more discerning about the value of particular studies and the influence on analytical decision. This issue will become more important as the number of primary qualitative research studies grows. Although our reviewers agreed about which studies were 'key', 'fatally flawed' or 'irrelevant'[18] the majority of studies were appraised as 'satisfactory'. As only five studies were appraised as 'key' this status did not influence data analysis. Secondly, GRADE CERQual considers adequacy (weight) and coherence (consistency) of data as important factors that can contribute to confidence in a review finding. However do these necessarily equate to validity and how do we know what is adequate? The issue of determining adequacy resonates with the unresolved question 'how many qualitative interviews is enough?'[74]. We chose to rate our confidence in a finding as high when a theme was supported by a least half of the studies ($n \geq 39$). However, although you could argue that weight and consistency [75] of findings contribute to the persuasiveness of a finding, it is important to consider that a unique idea can exert a significant pull. It is thus important not to ignore unique or inconsistent findings.

27. The use of HCP as a term ultimately rings a bit hollow for me in the discussion with the reference to prescribing of opioids which positions the main meaning of HCP as a medical role i.e. the doctor as opposed to other professions.

We have now included participant details in response to both reviewers in table 1 and for exemplary quotes.

28. Overall I feel that the paper has merits in its efforts to assimilate literature on this critically important topic. However I wonder if the stage presented by a set of 77 studies is being oversimplified or to what degree the meta ethnography is the most appropriate strategy – I would like to have this explained and argued more strongly.

We are pleased that you feel the paper has merits. We present the only review of HCPs experience of treating chronic pain and offer a novel line of argument that we feel will help HCPs to critically consider their approach to chronic pain. We are confident in the use of meta-ethnography for a large QES and have experience and previous publication in this area. We direct readers to our methodological paper for large meta-ethnographies.

We will use the methods of meta-ethnography developed, refined and reported in a previous meta-ethnography of patients' experience of chronic musculoskeletal pain.[12]

29. I find the presentation of the findings does include conceptual organisation yet what is being done in the presentation of the individual points and use of quotes strikes me as simplistic, very little attention to nuances of language use in the quotes.

We do not focus on the nuances of language because the data of analysis for meta-ethnography is not the primary first-order narrative. Rather it is the second-order findings of each primary study. Schütz distinction between (1) first-order constructs (the participants' 'common sense' interpretations in their own words) and (2) second-order constructs (the researchers' interpretations based on first-order constructs) is a useful one. The 'data' of meta-ethnography are second-order constructs. In meta-ethnography, these second-order constructs are then further abstracted to develop third-order constructs (the researchers' interpretations of the original authors' interpretations).

30. I am also reiterating the issue about the diversity of contexts, disciplines, etc. and how this is not at all apparent when the authors present their synthesis.

We have now addressed this issue in the revised table 1.

I find some aspects of the discussion disheartening – the centrality of the practitioner as the determiner of the care process and reference to 'I' too closely aligning authors with the practitioners, and little critical questioning of HCP role and power. Finally I don't feel the mixing desk analogy follows naturally from the findings and feels a bit bolted on.

You are correct that this study focuses centrally on practitioners as it is based on studies that explore their experience. Our previous studies and film have focused on patient experience <https://www.youtube.com/watch?v=FPpu7dXJFRI/>. Our analysis is a collaborative endeavour between social science, medical, allied and nursing professions and patients with chronic pain who played a central role in analysis.

We have added:

Now we have a body of qualitative knowledge exploring patients' experiences of chronic pain[76] and healthcare professionals experiences, the next challenge in practice is to bring these two bodies of knowledge together and look at how HCPs and patients can work together in managing pain.

1. IASP: Classification of Chronic Pain. Seattle, USA: International Association for the Study of Pain (IASP) Press; 1994.

2. Donaldson L: 2009 Annual Report of the Chief Medical Office (Department of Health. In. http://www.sthc.co.uk/Documents/CMO_Report_2009.pdf [accessed 11 Sept 12]; 2009.

3. Fayaz A, Croft P, Langford RM, Donaldson LJ, Jones GT: Prevalence of chronic pain in the UK: a systematic review and meta-analysis of population studies. *BMJ Open* 2016, 6(6):<http://bmjopen.bmj.com/content/6/6/e010364>.

4. Breivik H, Collett B, Ventafridda V, Cohen R, Gallacher D: Survey of chronic pain in Europe: Prevalence, impact on daily life, and treatment. *European Journal of Pain* 2006, 10(4):287-333.

5. Nahin R: Estimates of Pain Prevalence and Severity in Adults: United States, 2012. *Journal of Pain* 2015, 16(8):769-780.
6. Price C, Hoggart B, Olukoga O, CdeC-Williams A, Bottle A: National Pain Audit: British Pain Society; 2012.
7. Noblit G, Hare R: *Meta-ethnography: Synthesising Qualitative Studies*. California: Sage Publications; 1988.
8. Britten N, Campbell R, Pope C, Donovan J, Morgan M, Pill R: Using meta ethnography to synthesise qualitative research: a worked example. *Journal of Health Services and Research Policy* 2002, 7(4):209-215.
9. Campbell R, Pound P, Pope C, Britten N, Pill R, Morgan M, Donovan J: Evaluating meta-ethnography: a synthesis of qualitative research on lay experience of diabetes and diabetes care. *Social Science & Medicine* 2003, 56:671-684.
10. Malpass A, Shaw A, Sharp D, Walter F, Feder G, Ridd M, Kessler D: "Medication career" or "Moral career"? The two sides of managing anti-depressants: A meta-ethnography of patients experience of antidepressants. *Social Science and Medicine* 2009, 68:154-168.
11. Barker K, Toye F, MinnsLowe C: A qualitative systematic review of patients' experience of osteoporosis using meta-ethnography. *Archives of Osteoporosis* 2016, 11(33):DOI 10.1007/s11657-11016-10286-z.
12. Toye F, Seers K, Allcock N, Briggs M, Carr E, Andrews J, Barker K: A meta-ethnography of patients' experiences of chronic non-malignant musculoskeletal pain. *Health Services and Delivery Research* 2013, 1(12):1-189.
13. Toye F, Seers K, Allcock N, Briggs M, Carr E, Barker K: A synthesis of qualitative research exploring the barriers to staying in work with chronic musculoskeletal pain. *Disability & Rehabilitation* 2016, 38(6):566-572.
14. Toye F, Seers K, Barker K: A meta-ethnography of patients' experiences of chronic pelvic pain: struggling to construct chronic pelvic pain as 'real'. *Journal of Advanced Nursing* 2014, 70(12):2713-2727.
15. Toye F, Seers K, Allcock N, Briggs M, Carr E, Barker K: Meta-ethnography 25 years on: challenges and insights for synthesising a large number of qualitative studies. *BMC Medical Research Methodology* 2014, 14(80).
16. Sandelowski M, Barroso J: *Handbook for synthesising qualitative research*. New York: Springer Publishing Company; 2007.
17. Dixon-Woods M, Booth A, Sutton A: Synthesizing qualitative research: a review of published reports. *Qualitative Research* 2007, 7:375-422.
18. Dixon-Woods M, Agarwal S, Jones D, Young B, Sutton A: Synthesising qualitative and quantitative research evidence: a review of possible methods. *Journal of Health Services and Research Policy* 2005, 10(1):45-53.
19. Hannes K, Macaitis K: A move to more systematic and transparent approaches in qualitative evidence synthesis: update on a review of published papers. *Qualitative Research* 2012, 12 (4):402-442
20. Barnett-Page E, Thomas J: *Methods for synthesis of qualitative research: a critical review*. Economic and Social Research Council Research Methods 2009, National Centre for Research Methods Working Paper Series(01/09).
21. Pope C, Mays N, Popay J: *Synthesizing Qualitative and Quantitative Health Research: A Guide to Methods*. Berkshire, UK: Open University Press; 2007.
22. Toye F, Seers K, Allcock N, Briggs M, Carr E, Andrews J, Barker K: Patients' experiences of chronic non-malignant musculoskeletal pain: a qualitative systematic review. *The British journal of general practice : the journal of the Royal College of General Practitioners* 2013, 63(617).
23. Nvivo: *Nvivo qualitative data analysis and software*, QSR International Pty Ltd. In: software for qualitative data analysis. 9 edn; 2010.
24. Frost J, Garside R, Cooper C, Britten N: Meta-study as diagnostic: toward content over form in qualitative synthesis. *Qualitative Health Research* 2016, 26(3):307-3019.

25. Helman C: Culture Health and Illness 4th edn: Butterworth Heinemann. ; 2007.
26. Åsbring P, Närvänen A: Ideal versus reality: physicians perspectives on patients with chronic fatigue syndrome (CFS) and fibromyalgia. *Social Science & Medicine* 2003, 57(4):711-721.
27. Barry T, Irwin S, Jones S, Becker C, Tetrault M, Sullivan E, Hansen H, O'Connor G, Schottenfeld S, Fiellin A: Opioids, chronic pain, and addiction in primary care. *The journal of pain : official journal of the American Pain Society* 2010, 11(12):1442.
28. Bergman A, Matthias S, Coffing M, Krebs E: Contrasting Tensions Between Patients and PCPs in Chronic Pain Management: A Qualitative Study. *Pain Medicine* 2013, 14(11):1689-1698.
29. Blomqvist K: Older people in persistent pain: nursing and paramedical staff perceptions and pain management. *Journal of Advanced Nursing* 2003, 41(6):575-585.
30. Clark L, Jones K, Pennington K: Pain assessment practices with nursing home residents. *Western Journal of Nursing Research* 2004, 26(7):733-751.
31. Clark L, Fink R, Pennington K, Jones K: Nurses' Reflections on Pain Management in a Nursing Home Setting. *Pain Management Nursing* 2006, 7(2):71.
32. Côté P, Clarke J, Deguire S, Frank JW, Yassi A: Chiropractors and return-to-work: the experiences of three Canadian focus groups. *Journal of manipulative and physiological therapeutics* 2001, 24(5):309.
33. Dahan R, Borkan J, Brown JB, Reis S, Hermoni D, Harris S: The challenge of using the low back pain guidelines: a qualitative research. *Journal of Evaluation in Clinical Practice* 2007, 13(4):616-620.
34. Daykin AR, Richardson B: Physiotherapists' pain beliefs and their influence on the management of patients with low back pain. *Spine* 2004, 29(7):783-795.
35. Dobbs D, Baker T, Carrion IV, Vongxaiburana E, Hyer K: Certified nursing assistants' perspectives of nursing home residents' pain experience: communication patterns, cultural context, and the role of empathy. *Pain management nursing : official journal of the American Society of Pain Management Nurses* 2014, 15(1):87-96.
36. Eccleston C, De CACA, Rogers WS: Patients' and professionals' understandings of the causes of chronic pain: blame, responsibility and identity protection. *Social Science & Medicine* 1997, 45(5):699-710.
37. Espeland A, Baerheim A: Factors affecting general practitioners' decisions about plain radiography for back pain: implications for classification of guideline barriers--a qualitative study. *BMC Health Services Research* 2003, 3(1).
38. Esquibel AY, Borkan J: Doctors and patients in pain: Conflict and collaboration in opioid prescription in primary care. *Pain* 2014, 155(12):2575-2582.
39. Hayes M, Myhal C, Thornton F, Camerlain M, Jamison C, Cytryn N, Murray S: Fibromyalgia and the therapeutic relationship: Where uncertainty meets attitude. *Pain Research & Management* 2010, 15(6):385-391.
40. Hellstrom O, Bullington J, Karlsson G, Lindqvist P, Mattsson B: Doctors' attitudes to fibromyalgia: a phenomenological study. *Scandinavian Journal of Social Medicine* 1998, 26(3):232-237.
41. Holloway K, McConigley R: Descriptive, exploratory study of the role of nursing assistants in Australian residential aged care facilities: The example of pain management. *Australasian Journal on Ageing* 2009, 28(2):70-74.
42. Kaasalainen S, Brazil K, Coker E, Ploeg J, Martin-Misener R, Donald F, DiCenso A, Hadjistavropoulos T, Dolovich L, Papaioannou A et al: An action-based approach to improving pain management in long-term care. *Canadian Journal on Aging* 2010, 29(4):503-518.
43. Krebs E, Bergman A, Coffing M, Campbell R, Frankel M, Matthias S: Barriers to guideline-concordant opioid management in primary care—A qualitative study. *The Journal of Pain* 2014, 15(11):1148.
44. Kristiansson MH, Brorsson A, Wachtler C, Troein M: Pain, power and patience--a narrative study of general practitioners' relations with chronic pain patients. *BMC family practice* 2011, 12:31.
45. Lundh C, Segesten K, Björkelund C: To be a helpless helpoholic--GPs' experiences of women patients with non-specific muscular pain. *Scandinavian journal of primary health care* 2004, 22(4):244.

46. Macneela P, Gibbons A, McGuire B, Murphy A: "We need to get you focused": general practitioners' representations of chronic low back pain patients. *Qualitative Health Research* 2010, 20(7):977-986.
47. McCrorie C, Closs SJ, House A, Petty D, Ziegler L, Glidewell L, West R, Foy R: Understanding long-term opioid prescribing for non-cancer pain in primary care: a qualitative study. *BMC family practice* 2015, 16:121.
48. Parsons S, Harding G, Breen A, Foster N, Pincus T, Vogel S, Underwood M: Will shared decision making between patients with chronic musculoskeletal pain and physiotherapists, osteopaths and chiropractors improve patient care? *Family Practice* 2012, 29(2):203-212.
49. Siedlecki L, Modic MB, Bernhofer E, Sorrell J, Strumble P, Kato I: Exploring how bedside nurses care for patients with chronic pain: a grounded theory study. *Pain management nursing : official journal of the American Society of Pain Management Nurses* 2014, 15(3):565.
50. Spitz A, Moore A, Papaleontiou M, Granieri E, Turner J, Reid MC: Primary care providers' perspective on prescribing opioids to older adults with chronic non-cancer pain: a qualitative study. *BMC geriatrics* 2011, 11:35.
51. Starrels L, Wu B, Peyser D, Fox D, Batchelder A, Barg K, Arnsten H, Cunningham O: It made my life a little easier: primary care providers' beliefs and attitudes about using opioid treatment agreements. *Journal of opioid management* 2014, 10(2):95.
52. Thunberg KA, Carlsson SG, Hallberg LRMRM: Health care professionals' understanding of chronic pain: A grounded theory study. *Scandinavian journal of caring sciences* 2001, 15(1):99.
53. Toye F, Jenkins S, Seers K, Barker K: Exploring the value of qualitative research films in clinical education. *BMC Medical Research Methodology* 2015, 15(214):<http://bmcmmededuc.biomedcentral.com/articles/10.1186/s12909-12015-10491-12902>.
54. Stinson J, White M, Isaac L, Campbell F, Brown S, Ruskin D, Gordon A, Galonski M, Pink L, Buckley N et al: Understanding the information and service needs of young adults with chronic pain: perspectives of young adults and their providers. *The Clinical journal of pain* 2013, 29(7):600.
55. Wilson N, Pope C, Roberts L, Crouch R: Governing healthcare: Finding meaning in a clinical practice guideline for the management of non-specific low back pain. *Social Science & Medicine* 2014, 102:138.
56. Slade SC, Molloy E, Keating JL: The dilemma of diagnostic uncertainty when treating people with chronic low back pain: a qualitative study. *Clinical Rehabilitation* 2012, 26(6):558-570.
57. Afrell M, Rudebeck CE: 'We got the whole story all at once': physiotherapists' use of key questions when meeting patients with long-standing pain. *Scandinavian journal of caring sciences* 2010, 24(2):281.
58. Barker KL, Heelas L, Toye F: Introducing Acceptance and Commitment Therapy to a physiotherapy led pain rehabilitation programme: An Action Research Study. *British Journal of Pain* 2015.
59. Scott-Dempster C, Toye F, Truman J, Barker K: Physiotherapists' experiences of activity pacing with people with chronic musculoskeletal pain. An interpretative phenomenological analysis. *Physiotherapy Theory & Practice* 2014, 30(5):319-328.
60. Baszanger I: Deciphering chronic pain. *Sociology of Health and Illness* 1992, 14(2):181-215.
61. Cartmill C, Soklaridis S, Cassidy J: Transdisciplinary teamwork: The experience of clinicians at a functional restoration program. *Journal of occupational rehabilitation* 2011, 21(1):1.
62. Howarth M, Warne T, Haigh C: "Let's stick together" – A grounded theory exploration of interprofessional working used to provide person centered chronic back pain services. *Journal of interprofessional care* 2012, 26(6):491.
63. O'Connor BB, Eisenberg DM, Buring JE, Liang CL, Osypiuk K, Levy DB, Wayne PM: Within-team patterns of communication and referral in multimodal treatment of chronic low back pain patients by an integrative care team. *Global Advances In Health and Medicine* 2015, 4(2):36-45.

64. Oosterhof B, Dekker JH, Sloots M, Bartels EA, Dekker J: Success or failure of chronic pain rehabilitation: the importance of good interaction - a qualitative study under patients and professionals. *Disability and Rehabilitation* 2014, 36(22):1903-1910.
65. Sloots M, Scheppers EF, Bartels EAC, Dekker JHM, Geertzen JHB, Dekker J: First rehabilitation consultation in patients of non-native origin: factors that lead to tension in the patient-physician interaction. *Disability & Rehabilitation* 2009, 31(22):1853-1862.
66. Sloots M, Dekker JHM, Pont M, Bartels EA, Geertzen JHB, Dekker J: Reasons for drop-out from rehabilitation in patients of Turkish and Moroccan origin with chronic low back pain in The Netherlands: a qualitative. *Journal of Rehabilitation Medicine* 2010, 42(6):566-574.
67. Tveiten S, Meyer I: 'Easier said than done': empowering dialogues with patients at the pain clinic - the health professionals' perspective. *Journal of Nursing Management* 2009, 17(7):804-812.
68. Zanini C, Sarzi-Puttini P, Atzeni F, Di FM, Rubinelli S: Doctors' insights into the patient perspective: a qualitative study in the field of chronic pain. *BioMed Research International* 2014:514230-514231.
69. Scott-Dempster C, Toye F, Truman J, Barker K: Physiotherapists' experiences of activity pacing with people with chronic musculoskeletal pain: an interpretative phenomenological analysis. *Physiotherapy theory and practice* 2014, 30(5):319.
70. Briggs E, Carr E, Whittaker M: Survey of undergraduate pain curricula for healthcare professionals in the United Kingdom. *European Journal of Pain* 2011, 15(8):789-795.
71. Toye F, Seers K, Allcock N, Briggs M, Carr E, Andrews J, Barker K: 'Trying to pin down jelly' - exploring intuitive processes in quality assessment for meta-ethnography. *BMC Medical Research Methodology* 2013, 13:46.
72. Dixon-Woods M, Sutton A, Shaw R, Miller T, Smith J, Young B, Bonas S, Booth A, Jones D: Appraising qualitative research for inclusion in systematic reviews: a quantitative and qualitative comparison of three methods. *Journal of Health Services and Research Policy* 2007, 12(1):42-47.
73. Campbell R, Pound P, Morgan M, Daker-White G, Britten N, Pill R, Yardley L, Pope C, Donovan J: Evaluating meta-ethnography: systematic analysis and synthesis of qualitative research. *Health Technology Assessment* 2011, 15(43).
74. Baker SE, Edwards R: How many qualitative interviews is enough? National Centre for Research Methods Review Paper: Expert voices and early career reflections on sampling and cases in qualitative research 2012, <http://eprints.ncrm.ac.uk/2273/> (last accessed 12th July 2013).
75. Lewin S, Glenton C, Munthe-Kaas H, Carlsen B, Colvin CJ, Gülmezoglu M, Noyes J, Booth A, Garside R, Rashidian A: Using Qualitative Evidence in Decision Making for Health and Social Interventions: An Approach to Assess Confidence in Findings from Qualitative Evidence Syntheses (GRADE-CERQual). *PLOS Medicine* 2016, 12(10):e1001895. doi:1001810.1001371/journal.pmed.1001895.
76. Toye F, Seers K, Hannink E, Barker K: A mega-ethnography of eleven qualitative evidence syntheses exploring the experience of living with chronic non-malignant pain. *BMC Medical Research Methodology* 2017, 17(116):<https://bmcmedresmethodol.biomedcentral.com/track/pdf/10.1186/s12874-12017-10392-12877?site=bmcmedresmethodol.biomedcentral.com>.
77. Wong SS WN, Haynes RB. : Developing optimal search strategies for detecting clinically relevant qualitative studies in MEDLINE. *Medinfo* 2004, 11(1):311-316.
78. Wilczynski NL MS, Haynes RB. : Search strategies for identifying qualitative studies in CINAHL. *Qualitative Health Research* 2007, 17(5):705-710.
79. McKibbin KA WN, Haynes RB. : Developing optimal search strategies for retrieving qualitative studies in PsycINFO. *Evaluation & the Health Professions* 2006, 29(4):440-454.
80. Walters LA WN, Haynes RB; Hedges Team. : Developing optimal search strategies for retrieving clinically relevant qualitative studies in EMBASE. *Qualitative Health Research* 2006, 16(1):162-168.

81. Patel S, Peacock S, McKinley R, Carter DC, Watson PJ: GPs' experience of managing chronic pain in a South Asian community--a qualitative study of the consultation process. *Family practice* 2008, 25(2):71.
82. Patel S, Peacock SM, McKinley RK, Clark-Carter D, Watson PJ: GPs' perceptions of the service needs of South Asian people with chronic pain: A qualitative enquiry. *Journal of Health Psychology* 2009, 14(7):909-918.
83. Sloots M, Dekker HM, Pont M, Bartels AC, Geertzen HB, Dekker J: Reasons of drop-out from rehabilitation in patients of Turkish and Moroccan origin with chronic low back pain in The Netherlands: a qualitative study. *Journal of rehabilitation medicine* 2010, 42(6):566.
84. Paulson M, Danielson E, Norberg A: Nurses' and physicians' narratives about long-term non-malignant pain among men. *Journal of Advanced Nursing* 1999, 30(5):1097-1105.
85. Cameron A, Smith H, Schofield A: HEALTH CARE PROFESSIONALS' ACCOUNTS OF CHRONIC PAIN MANAGEMENT FOR OLDER ADULTS. *Journal of the Physiotherapy Pain Association* 2015(38):17-28.
86. Fox P, Solomon P, Raina P, Jadad A: Barriers and facilitators in pain management in long-term care institutions: a qualitative study. *Canadian Journal on Aging* 2004, 23(3):269-280.
87. Kaasalainen S, Coker E, Dolovich L, Papaioannou A, Hadjistavropoulos T, Emili A, Ploeg J: Pain management decision making among long-term care physicians and nurses. *Western journal of nursing research* 2007, 29(5):561.
88. Löckenhoff E, Laucks S, Port D, Tung J, Wethington E, Reid MC: Temporal horizons in pain management: understanding the perspectives of physicians, physical therapists, and their middle-aged and older adult patients. *The Gerontologist* 2013, 53(5):850.
89. Mentes C, Teer J, Cadogan P: The pain experience of cognitively impaired nursing home residents: perceptions of family members and certified nursing assistants. *Pain management nursing : official journal of the American Society of Pain Management Nurses* 2004, 5(3):118.
90. Ruiz JG, Qadri SS, Nader S, Wang J, Lawler T, Hagenlocker B, Roos BA: Primary care management of chronic nonmalignant pain in veterans: a qualitative study. *Educational Gerontology* 2010, 36(5):372-394.
91. Blomberg A-M, Hylander I, Törnkvist L: District nurses' involvement in pain care: A theoretical model. *Journal of clinical nursing* 2008, 17(15):2022